# SeeDNorm: Self-Rescaled Dynamic Normalization

**Wenrui Cai**$^{\diamond,\spadesuit,*}$, **Defa Zhu**$^{\spadesuit,*,\boxtimes,\dagger}$, **Siyuan Qiao**$^{\spadesuit}$, **Qingjie Liu**$^{\diamond,\boxtimes}$, **Qiyang Min**$^{\spadesuit}$

$^{\spadesuit}$Bytedance Seed China, $^{\diamond}$School of Computer Science and Engineering, Beihang University

{wenrui_cai,qingjie.liu}@buaa.edu.cn

{zhudefa,siyuan.qiao,minqiyang}@bytedance.com

## Abstract

Normalization layer constitutes an essential component in neural networks. In transformers, the predominantly used RMSNorm constrains vectors to a unit hypersphere, followed by dimension-wise rescaling through a learnable scaling coefficient $\gamma$ to maintain the representational capacity of the model. However, RMSNorm discards the input norm information in forward pass and a static scaling factor $\gamma$ may be insufficient to accommodate the wide variability of input data and distributional shifts, thereby limiting further performance improvements, particularly in zero-shot scenarios that large language models routinely encounter. To address this limitation, we propose SeeDNorm, which enhances the representational capability of the model by dynamically adjusting the scaling coefficient based on the current input, thereby preserving the input norm information and enabling data-dependent, self-rescaled dynamic normalization. During backpropagation, SeeDNorm retains the ability of RMSNorm to dynamically adjust gradient according to the input norm. We provide a detailed analysis of the training optimization for SeeDNorm and proposed corresponding solutions to address potential instability issues that may arise when applying SeeDNorm. We validate the effectiveness of SeeDNorm across models of varying sizes in large language model pre-training as well as supervised and unsupervised computer vision tasks. By introducing a minimal number of parameters and with negligible impact on model efficiency, SeeDNorm achieves consistently superior performance compared to previously commonly used normalization layers such as RMSNorm and LayerNorm, as well as element-wise activation alternatives to normalization layers like DyT.

## 1 Introduction

Normalization layers have become a fundamental building block in modern deep neural networks, playing a key role in stabilizing training and accelerating convergence (Ioffe & Szegedy, 2015). By enforcing statistical regularity on activations, normalization techniques help prevent issues such as exploding or vanishing gradients, thereby enabling deeper and more expressive models. Over the past decade, normalization has proven indispensable, especially in large-scale architectures for both language modeling (Vaswani et al., 2017) and computer vision (He et al., 2016) fields.

However, this stability comes at a cost: conventional normalization methods (such as LayerNorm (Ba et al., 2016) and RMSNorm (Zhang & Sennrich, 2019)) tend to discard or diminish information about the input norm, which can restrict the expressive capacity of the network and hinder the preservation of crucial scale-related features. Although modern normalization layers introduce learnable parameters to restore some network expressivity, these parameters are static and input-independent, which presents challenges in scenarios such as zero-shot generalization.

Alternatively, saturation activation functions such as $\tanh$ or its dynamic variants (Zhu et al., 2025b) offer the potential to retain norm information by constraining outputs within a fixed range. While these functions can preserve the relative scale or norm of the input, they inevitably suffer from the vanishing gradient problem in extreme cases, and these functions are unable to dynamically adjust

---

$^{*}$ Equal Contribution; $^{\dagger}$ Project Lead; $^{\boxtimes}$ Correspondence to: Qingjie Liu and Defa Zhu.

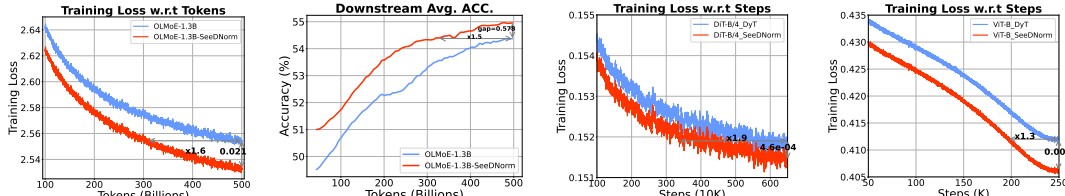

Figure 1: Comparisons between SeeDNorm and prior methods across diverse tasks in language modeling and computer vision. The first two figures depict training loss curve comparisons and average downstream task[1] accuracy between the OLMoE-1.3B (Muennighoff et al., 2024) baseline (using RMSNorm (Zhang & Sennrich, 2019)) and the SeeDNorm-equipped model, training on 500B tokens. The latter two figures show training loss comparisons between DyT-based (Zhu et al., 2025b) baseline models and SeeDNorm-based models in image generation and MAE (He et al., 2022) pre-training. All loss curves are smoothed with a 0.99 EMA.

gradients based on input norm during backpropagation like RMSNorm (Zhang & Sennrich, 2019), which leads to inefficient optimization and slow convergence.

This dilemma raises a fundamental question: *Is it possible to design a method that combines the training stability, the optimization efficiency, and the ability to preserve input norm information?*

In this work, we answer this question affirmatively by introducing **Self-Rescaled Dynamic Normalization (SeeDNorm)**. SeeDNorm achieves stable and efficient training while explicitly retaining norm information throughout the network. Extensive experiments demonstrate that SeeDNorm consistently accelerates convergence and improves performance across both language modeling and vision tasks, offering a simple yet effective alternative to existing normalization and activation approaches. As illustrated in Figure 1, integrating SeeDNorm into language and vision models leads to faster convergence and consistently improved performance across a variety of downstream tasks. In summary, our main contributions are as follows:

- We propose **SeeDNorm**, a dynamic normalization layer that generalizes RMSNorm and adaptively adjusts its scaling coefficient based on the current input, preserving the input norm information and offering improved adaptability to data variability and distributional shifts.

- We conduct a detailed and comprehensive analysis of the forward pass and gradients in the backpropagation of SeeDNorm, demonstrating the advantages of our method over existing normalization and dynamic activation alternatives, while also proposing techniques to enhance training stability.

- Extensive experiments on large language models with both dense and MoE (Du et al., 2022) structure, as well as computer vision tasks, show that SeeDNorm consistently accelerates convergence and improves performance over RMSNorm, LayerNorm and DyT baselines, with minimal additional parameters and computational cost.

We believe SeeDNorm offers a simple yet effective path towards more robust and flexible normalization in large-scale neural networks, especially for scenarios requiring strong generalization across diverse or shifting data distributions.

## 2 RELATED WORK

**Normalization layers.** Normalization layers are widely employed in modern deep neural networks, and the formulation of most normalization layers can be described as:

$$\text{Norm}(\boldsymbol{x}) = \boldsymbol{\gamma}\frac{\boldsymbol{x} - \boldsymbol{\mu}}{\sqrt{\boldsymbol{\sigma}^2 + \epsilon}} + \boldsymbol{\beta}. \quad (1)$$

$\epsilon$ is an extremely small value to prevent division by zero. Given $\boldsymbol{x} \in \mathbb{R}^{B \times N \times D}$, normalization layers enforce to transform the distribution over specific dimensions of the input to a standard normal

---

[1]All downstream tasks are evaluated in zero-shot manner, all tasks are shown in Table 5

distribution, ensuring that data distributions across network layers remain stable during training and accelerating model convergence. Subsequently, learnable parameters $\gamma$ and $\beta$ are used to adjust the distributions of each layer, enabling diversity in the distributions across network layers and alleviating the degradation of the expressive capacity.

Batch normalization (BN) (Ioffe & Szegedy, 2015) operates by normalizing all elements within each channel across the batch dimension, where the channel-specific mean $\mu$ and variance $\sigma$ are calculated as $\mu_c = \frac{1}{BN} \sum_{b,n} \boldsymbol{x}_{b,n,c}$ and $\sigma_c^2 = \frac{1}{BN} \sum_{b,n} (\boldsymbol{x}_{b,n,c} - \mu_c)^2$, respectively. BN has gained prominence in computer vision tasks, particularly in convolutional neural networks (He et al., 2016), due to its alignment with convolutional operations: identical kernels process features within the same channel across all spatial positions and batch samples. As discussed in (Santurkar et al., 2018), BN effectively smooths the loss landscape and enables more stable training of the model. However, BN is not suitable for sequence modeling tasks and may leak context information across samples within a batch. As a result, BN is seldom adopted in large language models or generative models.

Layer Normalization (LN) (Ba et al., 2016) addresses the limitations of BN in sequence modeling. Instead, LN normalizes the input across the feature dimension for each individual sample, where $\boldsymbol{\mu}$ and $\boldsymbol{\sigma}^2$ in Eq equation 1 denote the mean and variance computed along the last dimension of $\boldsymbol{x}$. LN is widely adopted in language modeling and Transformer architectures due to its independence from batch size.

Root Mean Square Layer Normalization (RMSNorm) (Zhang & Sennrich, 2019) further simplifies LN by omitting the mean subtraction and normalizing only by the root mean square of the input:

$$\mathrm{RMSNorm}(\boldsymbol{x}) = \boldsymbol{\gamma} \odot \frac{\boldsymbol{x}}{\mathrm{RMS}(\boldsymbol{x})}, \texttt{where} \ \ \mathrm{RMS}(\boldsymbol{x}) = \sqrt{\frac{1}{D} \sum_{i=1}^{D} x_i^2 + \epsilon} \tag{2}$$

$\odot$ is element-wise multiplication along channel dimension. RMSNorm has been shown to provide stable training and competitive performance, especially in large-scale Transformer models (Touvron et al., 2023). Despite the effectiveness, a fundamental limitation shared by these normalization methods is that *stability is obtained by sacrificing information related to the scale of the input, which potentially restrict the network's expressive capacity.*

**Saturation activation functions.** Recent works have also attempted to use saturation activation functions to replace normalization layers. For instance, Zhu et al. (2025b) proposes using dynamic $\texttt{tanh}$ (DyT) to substitute normalization, which can be formulated as:

$$\mathrm{DyT}(\boldsymbol{x}) = \boldsymbol{\gamma} \odot \tanh(\alpha \boldsymbol{x}) + \boldsymbol{\beta} \tag{3}$$

The input-dependent statistics are replaced with activation function and learnable scalar parameter $\alpha$. The scaling coefficient $\boldsymbol{\gamma}$ and shift coefficient $\boldsymbol{\beta}$ is also preserved. DyT explicitly preserves the norm of the input vector $\boldsymbol{x}$ in the forward pass and constrains extreme values via $\tanh$, thereby mapping input vector **within** a hypersphere with a radius of $\sqrt{D}$ to enhance training stability. However, DyT exhibits a vanishing-gradient problem in its extreme regions. Since $\nabla_{\boldsymbol{x}} \mathrm{DyT}(\boldsymbol{x}) = \alpha \mathrm{sech}^2(\alpha \boldsymbol{x}) \cdot \mathrm{diag}(\boldsymbol{\gamma})$, when $\boldsymbol{\gamma}$ is too small, $\alpha$ is either too small or too large, or $\boldsymbol{x}$ is excessively large, the gradient tends to approach 0. Since $\mathrm{sech}^2(\boldsymbol{x})$ is a higher-order infinitesimal of $\frac{1}{\boldsymbol{x}}$, it still leads to the problem of gradient vanishing when backpropagating to the preceding layers. Furthermore, in Proposition B.1, we demonstrate that under the assumption of constant input norm, RMSNorm is equivalent to DyT in terms of gradient w.r.t $\boldsymbol{x}$, which also indicates that DyT lacks the ability of RMSNorm to adaptively adjust gradients based on input norm during backpropagation.

Motivated by the limitations of existing normalization layers and saturating activation functions, we present **SeeDNorm**. SeeDNorm incorporates input norm information during the forward pass and mitigates the gradient vanishing issue in backpropagation observed in DyT, while can also adjust gradients based on input like RMSNorm.

## 3 SELF-RESCALED DYNAMIC NORMALIZATION (SEEDNORM)

The core design of SeeDNorm lies in dynamically adjusting the scaling factor of normalization layer based on the input, while incorporating input norm information. Building upon RMSNorm, we

implement dynamic adjustment of the scaling parameter $\gamma$. Given the input $x \in \mathbb{R}^{N \times D}$, SeeDNorm can be formulated as follows:

$$\mathbf{SeeDNorm}(x) = [\sigma(x \cdot \beta^T) \cdot \alpha + \gamma] \odot \frac{x}{\mathrm{RMS}(x)}, \quad \text{where } \mathrm{RMS}(x) = \sqrt{\frac{1}{D}\sum_{i=1}^{D} x_i^2 + \epsilon} \qquad (4)$$

where $\gamma \in \mathbb{R}^{1 \times D}$ denotes the learnable scaling factor in RMSNorm, $\beta \in \mathbb{R}^{1 \times D}$ represents the self-rescaling parameter. SeeDNorm performs matrix multiplication between the input $x$ and $\beta^T$, then activates the result using the nonlinear function $\sigma$ to obtain an intermediate output $\sigma(x \cdot \beta^T) \in \mathbb{R}^{N \times 1}$. To further enhance the dynamic adjustment capability of SeeDNorm, the intermediate output is subsequently multiplied with another learnable parameter $\alpha \in \mathbb{R}^{1 \times D}$, producing an element-wise rescaling matrix $[\sigma(x \cdot \beta^T) \cdot \alpha] \in \mathbb{R}^{N \times D}$. This rescaling matrix is conditioned on $x$ itself and modulates the static scaling factor $\gamma$, thereby incorporating input norm information and endowing SeeDNorm with the ability to dynamically adjust the rescaling factor for diverse inputs. In our implementation, the $\sigma$ function is instantiated using the $\tanh$ activation, which inherently constrains the output range to $[-1, 1]$, ensuring that large outliers in the input $x$ do not exert a significant influence on the scaling coefficient.

SeeDNorm can be used to replace all normalization layers in current Transformer-based models, including QueryNorm and KeyNorm (Henry et al., 2020) that are commonly employed in the attention modules in LLMs. Algorithm 1 presents the PyTorch-style pseudocode for the SeeDNorm implementation. The initialization method of $\gamma$ is consistent with that of RMSNorm, using 1-initialization, while $\beta$ is initialized with zero. The initialization of $\alpha$ can be adjusted via hyperparameters. In the following sections, we will also discuss parameter initialization methods combined with the analysis of SeeDNorm.

During the training process, for the parameter $\gamma$, we maintain the same regularization scheme as the baseline model; otherwise, by default, we do not apply weight decay or other regularization techniques. For $\alpha$ and $\beta$, however, we apply regularization, which is more beneficial for model training, and alleviating overfitting.

## 3.1 ANALYSIS OF SEEDNORM

In this section, we present a detailed analysis of the forward and backward propagation of SeeDNorm. For forward propagation, our primary focus is on its scale invariance, specifically whether it can maintain numerical stability when the input scale or norm undergoes significant changes. As for translation invariance, it has already been demonstrated in RMSNorm that this does not impact model performance, so we will not pursue further analysis on this aspect. For backward propagation, we concentrate on the gradients of SeeDNorm with respect to each parameter $\alpha$, $\beta$, $\gamma$ and the input $x$.

**Invariance Analysis.** *While SeeDNorm does not exhibit the same strict scale invariance as RMSNorm, it demonstrates insensitivity to input scaling.*

Since $\epsilon$ is an extremely small value, for ease of notation, we will omit it by default in the subsequent discussion. For a given input $x \in \mathbb{R}^{1 \times D}$, when $x$ is scaled by multiplying a factor of $k$, SeeDNorm can be expressed as:

$$\left[\sigma(kx \cdot \beta^T) \cdot \alpha + \gamma\right] \odot \frac{kx}{\sqrt{\frac{1}{D}\sum_{d=1}^{D}(kx_d)^2}} = \left[\sigma(kx \cdot \beta^T) \cdot \alpha + \gamma\right] \odot \frac{x}{\sqrt{\frac{1}{D}\sum_{d=1}^{D}(x_d)^2}} \qquad (5)$$

Therefore, when $x$ is scaled by a factor of $k$, the only component in SeeDNorm that changes is the self-rescaling matrix, which transforms from $f(x) = [\sigma(x \cdot \beta^T) \cdot \alpha + \gamma]$ to $f(kx) = [\sigma(kx \cdot \beta^T) \cdot \alpha + \gamma]$. The derivative of $f$ with respect to $x$ is: $\nabla_x f = \mathrm{sech}^2(x \cdot \beta^T)(\alpha^T \cdot \beta) = (1 - \tanh^2(x \cdot \beta^T))(\alpha^T \cdot \beta)$.

For very large values of $x$, $\mathrm{sech}^2(x)$ approaches $0$ and $\nabla_x f$ approaches $0$, indicating that $f(x)$ undergoes minimal change. Conversely, as $x$ approaches $0$, $\nabla_x f$ reaches its maximum value $\alpha^T \cdot \beta$. Therefore, to preserve insensitivity of SeeDNorm to input scaling, we initialize $\beta$ to $0$ to make $\nabla_x f$ is initialized with $0$, and we also add weight decay to $\alpha$ and $\beta$. Additionally, when $x$ is close to $0$,

$f(\boldsymbol{x})$ is primarily dominated by $\boldsymbol{\gamma}$. Consequently, SeeDNorm is not significantly affected by the scale of the input magnitude.

**Gradient Analysis.** *Since SeeDNorm operates on each token individually, we assume by default that $\boldsymbol{x} \in \mathbb{R}^{1 \times D}$. For notational convenience, let $\boldsymbol{s} = \sigma(\boldsymbol{x} \cdot \boldsymbol{\beta}^T) \cdot \boldsymbol{\alpha}$. The gradients of the output of SeeDNorm with respect to $\boldsymbol{\alpha} \in \mathbb{R}^{1 \times D}$, $\boldsymbol{\beta} \in \mathbb{R}^{1 \times D}$, $\boldsymbol{\gamma} \in \mathbb{R}^{1 \times D}$ and $\boldsymbol{x}$ are as following:*

$$\frac{\partial \, \mathbf{SeeDNorm}(\boldsymbol{x})}{\partial \, \boldsymbol{\gamma}} = \mathrm{diag}(\frac{\boldsymbol{x}}{\mathrm{RMS}(\boldsymbol{x})})$$

$$\frac{\partial \, \mathbf{SeeDNorm}(\boldsymbol{x})}{\partial \, \boldsymbol{\alpha}} = \frac{\boldsymbol{x}}{\mathrm{RMS}(\boldsymbol{x})} \cdot \left[ \sigma(\boldsymbol{x} \cdot \boldsymbol{\beta}^T) \boldsymbol{I}_{D \times D} \right]$$

$$\frac{\partial \, \mathbf{SeeDNorm}(\boldsymbol{x})}{\partial \, \boldsymbol{\beta}} = \sigma'(\boldsymbol{x} \cdot \boldsymbol{\beta}^T) \left( \left( \boldsymbol{\alpha} \odot \frac{\boldsymbol{x}}{\mathrm{RMS}(\boldsymbol{x})} \right)^T \cdot \boldsymbol{x} \right) \tag{6}$$

$$\frac{\partial \, \mathbf{SeeDNorm}(\boldsymbol{x})}{\partial \, \boldsymbol{x}} = \sigma'(\boldsymbol{x} \cdot \boldsymbol{\beta}^T)(\boldsymbol{\alpha} \odot \frac{\boldsymbol{x}}{\mathrm{RMS}(\boldsymbol{x})})^T \cdot \boldsymbol{\beta} + \frac{1}{\mathrm{RMS}(\boldsymbol{x})} \left( \mathrm{diag}(\boldsymbol{s} + \boldsymbol{\gamma}) - \frac{(\boldsymbol{s} + \boldsymbol{\gamma})^T \mathbf{1}_{1 \times D}}{D \cdot \mathrm{RMS}^2(\boldsymbol{x})} \odot (\boldsymbol{x}^T \cdot \boldsymbol{x}) \right)$$

Detailed derivations of the gradients are provided in Appendix C.

**Gradient of $\boldsymbol{\gamma}$.** As demonstrated in Equation 5, the term $\frac{\boldsymbol{x}}{\mathrm{RMS}(\boldsymbol{x})}$ is scale-invariant. Therefore, the gradient of $\boldsymbol{\gamma}$ is not affected by samples of $\boldsymbol{x}$ that are abnormally large or abnormally small. This property contributes to the stability of the training of $\boldsymbol{\gamma}$.

**Gradient of $\boldsymbol{\alpha}$.** The gradient of $\boldsymbol{\alpha}$ also includes the scale-invariant term $\frac{\boldsymbol{x}}{\mathrm{RMS}(\boldsymbol{x})}$, but it is multiplied by $\sigma(\boldsymbol{x} \cdot \boldsymbol{\beta}^T)$. When $\sigma(\cdot)$ is implemented using $\tanh$, for abnormally large values of $\boldsymbol{x}$, the value can be constrained within 1, thereby preventing gradient explosion. If an abnormally small input occurs, the gradient of $\boldsymbol{\alpha}$ will similarly become small, but $\boldsymbol{\gamma}$ can still update normally. Additionally, since $\boldsymbol{\alpha}$ directly multiplies in the gradient of $\boldsymbol{\beta}$, and $\boldsymbol{\beta}$ also directly influences the gradient of $\boldsymbol{\alpha}$, $\boldsymbol{\alpha}$ and $\boldsymbol{\beta}$ cannot both be initialized to $\mathbf{0}$ simultaneously.

**Gradient of $\boldsymbol{\beta}$.** $\sigma'(\boldsymbol{x} \cdot \boldsymbol{\beta}^T) = \frac{1}{\cosh^2(\boldsymbol{x})}$. When $\boldsymbol{x}$ is abnormally large, $\cosh(\boldsymbol{x})$ is a higher-order infinitesimal of $\boldsymbol{x}$, so the gradient of $\boldsymbol{\beta}$ approaches 0. Similarly, when $\boldsymbol{x}$ is abnormally small, the gradient of $\boldsymbol{\beta}$ also approaches 0, thus avoiding the risk of gradient explosion. Since $\boldsymbol{\beta}$ is often encapsulated within $\sigma$ or $\sigma'$ in the gradient, which constrains its range, while $\boldsymbol{\alpha}$ is directly multiplied, we initialize $\boldsymbol{\beta}$ to zero, ensuring that the gradient of $\boldsymbol{\alpha}$ starts from 0 in the early stages of training, thereby enhancing training stability. At the same time, given that almost all gradients involve $\boldsymbol{\alpha}$ and $\boldsymbol{\beta}$, we need to control the scale of $\boldsymbol{\alpha}$ and $\boldsymbol{\beta}$ to prevent them from continuously increasing and causing excessively large gradients. Therefore, we apply weight decay to both parameters during the training process.

**Gradient of $\boldsymbol{x}$.** Unlike the gradients of other parameters, $\boldsymbol{x}$ is the activation output of the preceding layer, and its gradient is propagated to the previous layer for parameter updates. Therefore, when $\boldsymbol{x}$ is excessively large, SeeDNorm should output proportional small gradient w.r.t. $\boldsymbol{x}$, and vice versa. Assuming $\boldsymbol{x}$ is scaled to an abnormally large $k\boldsymbol{x}$, the first term approaches 0, and in the second term, $\boldsymbol{s}$ approaches 1, while $\frac{(k\boldsymbol{x})^T \cdot (k\boldsymbol{x})}{\mathrm{RMS}^2(k\boldsymbol{x})} = \frac{\boldsymbol{x}^T \cdot \boldsymbol{x}}{\mathrm{RMS}^2(\boldsymbol{x})}$ remains unchanged. Therefore, the gradient of $k\boldsymbol{x}$ is primarily dominated by $\frac{1}{\mathrm{RMS}(k\boldsymbol{x})} = \frac{1}{k\mathrm{RMS}(\boldsymbol{x})}$. Therefore, the gradient of SeeDNorm with respect to $k\boldsymbol{x}$ decreases by the same factor $k$. Thus, during backpropagation, SeeDNorm can dynamically adjust gradients based on the input norm like RMSNorm. Similarly, when $k\boldsymbol{x}$ is abnormally small, the second term is significantly larger than the first term and is also dominated by $\frac{1}{\mathrm{RMS}(k\boldsymbol{x})}$. Therefore, SeeDNorm exhibits favorable adaptive gradient adjustment properties during backpropagation. More analysis regarding gradients can refer to Appendix C.

## 3.2 MULTI-HEAD SEEDNORM

The $\sigma(\boldsymbol{x} \cdot \boldsymbol{\beta}^T)$ and $\sigma'(\boldsymbol{x} \cdot \boldsymbol{\beta}^T)$ term affects the gradients of $\boldsymbol{\alpha}$, $\boldsymbol{\beta}$, and $\boldsymbol{x}$. In our previous analysis, we focused only on extreme values, but in the actual training optimization process, such situations are rare. To further ensure training stability under non-extreme conditions, our strategy is to reduce the variance of this term, specifically, to decrease the variance of $\boldsymbol{x} \cdot \boldsymbol{\beta}^T$.

**Theorem 3.2.** *In high-dimensional space, the variance of the dot product of two random vectors is proportional to their dimension $D$.*

The proof of Theorem 3.2 can be found in the Appendix C.6. Therefore, we propose a multi-head form of SeeDNorm, which splits $x$ and $\beta$ into multiple sub-vectors and computes the dot product between these sub-vectors. This operation reduces the dimensionality of each dot product, and the results are then concatenated back to the original dimension, thereby achieving the goal of reducing variance. The process can be formally described as:

$$x = [x_{h_1}, x_{h_2}, ..., x_{h_n}], \ \beta = [\beta_{h_1}, \beta_{h_2}, ..., \beta_{h_n}]$$
$$\textbf{MHSeeDNorm} = \left[ \sigma \left( \left[ x_{h_1} \cdot \beta_{h_1}^T, ..., x_{h_n} \cdot \beta_{h_n}^T \right] \right) \cdot \alpha + \gamma \right] \odot \frac{x}{\text{RMS}(x)} \tag{7}$$

Under the multi-head form, the gradients of SeeDNorm with respect to each parameter and the input are also analyzed in detail in Appendix C. And Algorithm 2 is the pseudocode. The primary change is that $\sigma(\cdot)$ and $\sigma'(\cdot)$ are also transformed into a multi-head form, thereby achieving the goal of reducing gradient variance.

## 4 EXPERIMENTS

To validate the effectiveness and generality of SeeDNorm, we conduct comprehensive experiments across both language and vision tasks. During the experimental process, we systematically replace all normalization layers or saturation activation functions in the baseline models with our proposed SeeDNorm.

### 4.1 LARGE LANGUAGE MODELS

Our experiments on language modeling primarily focus on the pretraining of large language models. We conduct experiments under both dense and mixture-of-experts (MoE) (Shazeer et al., 2017; Fedus et al., 2022) model architectures. We selected OLMoE (Muennighoff et al., 2024) as the baseline for MoE architectures and OLMo2 (OLMo et al., 2024) as the baseline for dense model. To ensure experimental consistency, We utilize the identical training corpus *OLMoE-mix-0924* (Muennighoff et al., 2024) as specified in the original OLMoE implementation, and employ the *OLMo-mix-1124* (OLMo et al., 2024) dataset identical with OLMo2.

Both OLMoE and OLMo2 adopt RMSNorm as normalization layer. In addition to attention layers and FFNs, RMSNorm are also applied in output normalization, QueryNorm, and KeyNorm. In our experiments, we replace all normalization layers in the models with SeeDNorm, and perform QueryNorm and KeyNorm in each attention head. In the experiments of language modeling, the parameter $\alpha$ of SeeDNorm is initialized to $1$.

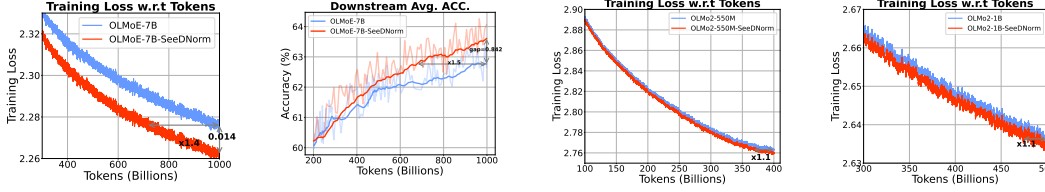

Figure 2: The first two subplots respectively show a comparison of training loss and average downstream task accuracy between OLMoE-7B baseline and the counterparts with SeeDNorm. The last two subplots show a comparison of training loss of OLMo2-550M and OLMo2-1B baseline models and their counterparts with SeeDNorm incorporated. All curves are smoothed using EMA with a coefficient of 0.99.

**Downstream Tasks and Evaluation Metrics.** Beyond evaluating each model based on the convergence behavior of the cross-entropy loss during training in Figure 2, we also report the average accuracy curves for the downstream tasks listed in Table 5. Table 1 presents several representative dataset metrics, we report the average perplexities (PPL) and losses on the *c4_en-validation* dataset, and simultaneously evaluate accuracy metrics on ARC-Challenge (Clark et al., 2018), ARC-Easy (Clark et al., 2018), HellaSwag (Zellers et al., 2019), MMLU-Var (Hendrycks et al.), and PIQA (Bisk et al., 2020).

Table 1: Performance comparison of dense models and MoE models of different sizes on the *c4_en-validation* dataset and multiple downstream datasets, where all metrics for downstream tasks are reported as **Acc. %**.

| Models | Training Tokens (B) | c4_en-validation | | Downstream Evaluation | | | | |
|---|---|---|---|---|---|---|---|---|
| | | Loss ↓ | PPL ↓ | ARC-C ↑ | ARC-E ↑ | HellaSwag ↑ | MMLU-Var ↑ | PIQA ↑ |
| *MoE Models* | | | | | | | | |
| OLMoE-1.3B | 500 | 2.922 | 18.63 | 32.3 | 62.2 | 55.2 | 32.4 | 72.6 |
| OLMoE-1.3B-DyT | 500 | 2.968 | 19.45 | 30.4 | 61.9 | 53.2 | 30.5 | 70.6 |
| OLMoE-1.3B-SeeDNorm | 500 | **2.900** | **18.12** | **34.5** | **65.4** | **56.8** | **33.2** | **73.1** |
| OLMoE-7B | 1000 | 2.644 | 14.07 | 40.8 | 73.7 | 71.2 | 38.8 | 76.6 |
| OLMoE-7B-SeeDNorm | 1000 | **2.631** | **13.88** | **44.5** | **76.1** | **71.8** | **40.2** | **79.1** |
| *Dense Model* | | | | | | | | |
| OLMo2-550M | 400 | 3.011 | 20.30 | 30.0 | 62.7 | **52.3** | 31.5 | 71.5 |
| OLMo2-550M-SeeDNorm | 400 | **3.008** | **20.24** | **31.4** | **63.4** | 52.0 | **31.6** | **71.5** |
| OLMo2-1B | 500 | 2.884 | 17.88 | 35.6 | 68.7 | 60.4 | 33.9 | 74.5 |
| OLMo2-1B-SeeDNorm | 500 | **2.879** | **17.79** | **37.8** | **70.0** | **61.0** | **34.8** | **74.5** |

**MoE Models.** We conduct experiments on OLMoE-1.3B and OLMoE-7B, with activated parameter counts of 260M and 1B, respectively. All experiments based on OLMoE-1.3B and OLMoE-7B are trained on 500B tokens and 1T tokens using the corresponding corpus datasets. Detailed configurations of additional experiments are provided in the Appendix D. As illustrated in Figure 1 and Figure 2, applying SeeDNorm to both the OLMoE-1.3B and OLMoE-7B models significantly accelerates the convergence during training. Furthermore, as the number of training tokens increases, models using SeeDNorm exhibit increasingly larger improvements in training loss compared to their baseline counterparts. Moreover, Table 1 and Figure 1 shows that both the 1.3B and 7B models achieve comprehensive improvements in various validation metrics and accuracy in downstream tasks. In contrast, replacing the normalization layers of the OLMoE-1.3B model with saturation activation function like DyT (Zhu et al., 2025b) leads to slow convergence and a degradation in performance.

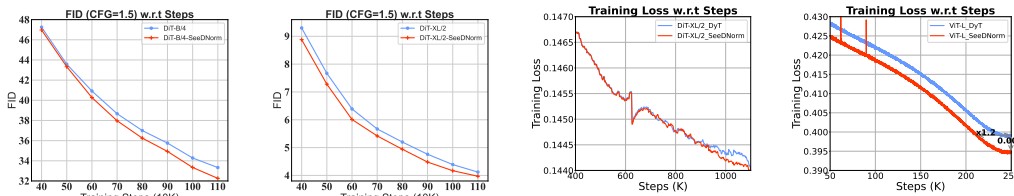

Figure 3: The first two subplots show FID comparison at different training steps wich CFG=1.5, and the last two subplots show loss curves of DiT-XL/2 in image generation and ViT-L in MAE. All models are respectively augmented with our proposed SeeDNorm and DyT.

**Dense Models.** Although MoE-based model architectures currently dominate LLMs, we also evaluate the performance of SeeDNorm in dense model. Our experiments are primarily conducted based on OLMo2-550M and OLMo2-1B, training on 400B and 500B tokens, respectively. Detailed configurations of additional experiments are provided in the Appendix D. As shown in Figure 2 and Table 1, SeeDNorm continues to yield benefits in dense models, with the training loss improvement over baselines still widening as the number of tokens increases. But the advantage of SeeDNorm in terms of training loss is reduced compared to MoE models. This may be because dense models do not require dynamic activation parameters, leading to more stable training, and each parameter is sufficiently trained, which diminishes the accelerated convergence advantage brought by SeeDNorm. However, in zero-shot evaluation tasks, such as ARC-C and ARC-E, the application of SeeDNorm can still significantly enhance performance. And the dynamic architectures of MoE models are better able to amplify the advantages of SeeDNorm.

## 4.2 COMPUTER VISION TASKS

We conducted experiments on supervised, unsupervised, and generative visual tasks, using the multi-head version of SeeDNorm across all tasks except image generation. More details are provided in Appendix D.

**Image Generation.** We evaluate the effectiveness of SeedNorm using Diffusion Transformer (DiT) (Peebles & Xie, 2023) as the baseline. Experiments are conducted on two model sizes: DiT-B/4 and DiT-XL/2, where 4 and 2 denote the patch sizes. It is noteworthy that SeeDNorm cannot directly replace AdaLN, the normalization layer within DiT. This limitation stems from the mechanism of AdaLN that incorporates class-specific information by predicting scaling parameter $\boldsymbol{\gamma}(c)$ and shifting parameter $\boldsymbol{\beta}(c)$ conditioned on class label $c$. Therefore, we retain the shift and scale terms of AdaLN, removed the $\boldsymbol{\gamma}$ inside SeeDNorm, and adopted the following form that includes label conditions:

$$\mathbf{AdaSeeDNorm}(\boldsymbol{x}, c) = \left[\left(\sigma(\boldsymbol{x} \cdot \boldsymbol{\beta^T}) \cdot \boldsymbol{\alpha} + 1\right) \odot \frac{\boldsymbol{x}}{\mathrm{RMS}(\boldsymbol{x})}\right](1 + \boldsymbol{\gamma}(c)) + \boldsymbol{\eta}(c), \text{ where } c \text{ is the condition} \tag{8}$$

We train DiT on the ImageNet-1K (Krizhevsky et al., 2012) dataset and evaluate on the ImageNet validation set, which comprises a total of 50,000 images. Since DyT (Zhu et al., 2025b) has already achieved better results than DiT baseline, we directly compare SeeDNorm with DyT in Figure 1 and Figure 3. We present comparisons of the loss curves and FID (Heusel et al., 2017) across different training steps. During evaluation, the cfg-scale is set to 1.5, consistent with the optimal value used for DiT (Peebles & Xie, 2023). Additional configuration details are provided in the Appendix D.4.

**Supervised Learning.** We conduct image classification experiments on two representative architectures, ViT (Dosovitskiy et al., 2021) and ConvNeXt (Liu et al., 2022). All models are trained on the ImageNet-1K (Krizhevsky et al., 2012) training set and evaluated on the test set. Additional configuration details are provided in the Appendix D.2. The results in Table 2 demonstrate that applying SeeDNorm achieves better performance compared to both DyT and LayerNorm.

Table 2: **Acc@1** on ImageNet-1K classification; (MAE) denotes fine-tuning MAE pretrained models.

| Model | LayerNorm | DyT | SeeDNorm |
|---|---|---|---|
| ViT-B | 82.3 | 82.5 | **82.7** |
| ViT-L | 83.1 | 83.6 | **83.6** |
| ConvNeXT-B | 83.7 | 83.7 | **83.7** |
| ConvNeXT-L | 84.3 | 84.4 | **84.6** |
| ViT-B (MAE) | 83.2 | 83.2 | **83.5** |
| ViT-L (MAE) | 85.5 | 85.4 | **85.5** |

**Self-Supervised Learning.** We select the representative self-supervised mask reconstruction task MAE (He et al., 2022) for experiments, which are conducted on ViT (Dosovitskiy et al., 2021). All models are initially pre-trained on the ImageNet-1K (Krizhevsky et al., 2012) dataset, and then fine-tuned on ImageNet-1K. Additional configuration details are provided in the Appendix D.3. Figures 1 and Figure 3 demonstrate that SeeDNorm significantly accelerates convergence during the pre-training stage, while Table 2 shows that it also holds an advantage during fine-tuning.

### 4.3 ABLATION STUDY

In our ablation studies presented in this section, we primarily focus on language modeling, conducting experiments with OLMoE-1.3B and OLMoE-7B as the baseline models. In the computer vision domain, we use ViT as the baseline model.

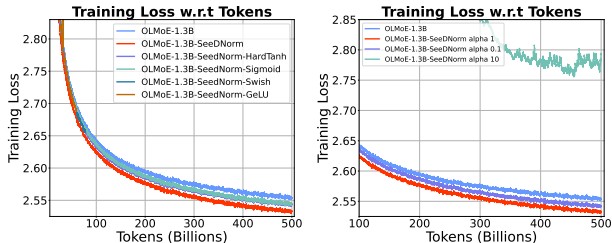

Figure 4: Two subplots present training loss curves of SeeDNorm with different activation functions, and with various $\boldsymbol{\alpha}$ initialization strategies, respectively. The loss curves are smoothed using EMA with a coefficient of 0.99.

**Implementation of Activation Function $\sigma$ in SeeDNorm.** In SeeDNorm, the activation function $\sigma$ is set to `tanh` by default. As demonstrated in Figure 4, we conduct experiments with different implementation choices for $\sigma$. When employing activation functions with an unbounded output range, such as GeLU (Hendrycks & Gimpel, 2016) and Swish (Ramachandran et al., 2017), the model fails to converge properly. In contrast, using bounded functions like `sigmoid`, `tanh`, and `hardtanh` allows the model to converge successfully. All bounded functions achieve performance superior to the baseline, with `tanh` yielding the best results.

Table 3: Ablation studies of SeeDNorm based on OLMoE. For OLMoE-1.3B, all experiments are trained on 500B tokens, while for OLMoE-7B, all experiments are trained on 1T tokens. We evaluate various models based on validation loss and PPL on the *c4_en-validation* dataset, and **Acc.%** on different downstream tasks. "←" denotes initialization, "$x \odot \beta$" indicates that $x$ and $\beta$ perform element-wise multiplication in SeeDNorm, "w/o" indicates removing the corresponding parameter, and "FC×$n$" indicates model with Dynamic Frac-Connections at $n$ frac-rates.

| Models | c4_en-validation | | Downstream Evaluation | | | | |
|---|---|---|---|---|---|---|---|
| | Loss ↓ | PPL ↓ | ARC-C ↑ | ARC-E↑ | HellaSwag↑ | MMLU-Var↑ | PIQA↑ |
| OLMoE-1.3B | 2.922 | 18.63 | 32.3 | 62.2 | 55.2 | 32.4 | 72.6 |
| OLMoE-1.3B-SeeDNorm ($\alpha \leftarrow 1$) | 2.900 | 18.12 | 34.5 | 65.4 | 56.8 | 33.2 | 73.1 |
| OLMoE-1.3B-SeeDNorm ($\alpha \leftarrow 0.1$) | 2.912 | 18.39 | 31.2 | 63.7 | 55.6 | 32.3 | 72.8 |
| OLMoE-1.3B-SeeDNorm ($\alpha \leftarrow 10$) | 3.154 | 23.42 | 27.8 | 53.0 | 43.0 | 28.6 | 68.2 |
| OLMoE-1.3B-SeeDNorm (scalar $\alpha$) | 2.909 | 18.33 | 32.6 | 62.2 | 55.9 | 32.4 | 72.6 |
| OLMoE-1.3B-SeeDNorm ($x \odot \beta$) | 2.909 | 18.33 | 36.5 | 64.9 | 55.7 | 32.0 | 72.7 |
| OLMoE-1.3B-SeeDNorm (w/o $\alpha$) | 2.907 | 18.29 | 32.1 | 67.0 | 56.5 | 32.9 | 73.2 |
| OLMoE-1.3B-SeeDNorm (w/o $\beta$) | 2.911 | 18.37 | 31.9 | 63.7 | 55.4 | 31.7 | 72.9 |
| OLMoE-1.3B-SeeDNorm (w/o $\gamma$) | 2.913 | 18.41 | 33.7 | 65.4 | 56.0 | 32.5 | 72.8 |
| OLMoE-1.3B-QKNormAll | 2.902 | 18.20 | 34.1 | 64.2 | 56.2 | 32.6 | 74.2 |
| OLMoE-1.3B-MultiheadSeeDNorm | 2.904 | 18.25 | 31.5 | 63.9 | 55.7 | 32.9 | 71.9 |
| OLMoE-1.3B-FC×2 | 2.908 | 18.32 | 32.1 | 63.5 | 55.9 | 31.6 | 72.4 |
| OLMoE-1.3B-SeeDNorm-FC×2 | 2.899 | 18.11 | 34.8 | 64.7 | 56.9 | 33.4 | 73.9 |
| OLMoE-7B | 2.644 | 14.07 | 40.8 | 73.7 | 71.2 | 38.8 | 76.6 |
| OLMoE-7B-SeeDNorm | 2.631 | 13.88 | 44.5 | 76.1 | 71.8 | 40.2 | 79.1 |
| OLMoE-7B-FC×4 | 2.630 | 13.88 | 44.3 | 75.6 | 71.9 | 39.6 | 78.2 |
| OLMoE-7B-SeeDNorm-FC×4 | 2.629 | 13.86 | 44.9 | 76.6 | 72.4 | 39.9 | 79.1 |

**Initialization of $\alpha$.** As depicted in Figure 4, experiments are conducted with $\alpha$ initialized to 0.1, 1, and 10, respectively. Figure 4 indicates that excessively large values of $\alpha$ adversely degrade training stability and lead to poorer final performance. In contrast, smaller initial $\alpha$ can maintain training stability, and a suitably larger initial $\alpha$ can accelerate model convergence.

**Whether $\alpha$ a vector or scalar.** In SeeDNorm, $\alpha$ is designed as a vector, which is used to generate an element-wise self-rescaling matrix of the same shape as the input $x$. Table 3 presents our evaluation of the performance impact when replacing $\alpha$ with a single scalar parameter. With $\alpha$ as a scalar, the self-rescaling matrix of SeeDNorm can only adjust each input vector uniformly, rather than providing element-specific scaling. As shown in Table 3, substituting the $D$-dimensional vector $\alpha$ with a scalar leads to a degradation in model performance, but it still surpasses the baseline.

**The type of multiplication between $\beta$ and $x$.** SeeDNorm uses the dot product between $\beta$ and $x$ by default. We further experimented with element-wise multiplication between $\beta$ and $x$. Although this does not affect the shape of the self-rescaling matrix, the results in Table 3 indicate that using the dot product method offers better expressive power and performance.

**Impact of Different Parameters in SeeD-Norm.** SeeDNorm incorporates three learnable parameters: $\alpha$, $\beta$, and $\gamma$, all of which are $D$-dimensional vectors. Table 3 presents an ablation study evaluating the impact on model performance when each of these parameters is removed. When $\alpha$ is removed, similar to the case where $\alpha$ is a scalar, the self-rescaling matrix is no longer element-wise; when $\beta$ is removed, SeeDNorm loses the ability to adjust the shape of the nonlinear function $\sigma$, and the computation with $\alpha$ becomes a matrix mul-

Table 4: Ablation Study on whether to apply weight decay to $\alpha$ and $\beta$ and the number of heads in SeeDNorm in ImageNet-1K classification.

| Model | Acc@1 | Acc@5 |
|---|---|---|
| ViT-B-SeeDNorm (16Head) | 82.7 | 96.1 |
| ViT-B-SeeDNorm (1Head) | Fail to converge | |
| ViT-B-SeeDNorm (8Head) | 82.5 | 96.0 |
| ViT-B-SeeDNorm (32Head) | 82.5 | 96.0 |
| ViT-B-SeeDNorm (*w/o* Weight Decay) | 82.4 | 95.9 |
| ViT-L-SeeDNorm (8Head) | 83.4 | 96.3 |
| ViT-L-SeeDNorm (32Head) | 83.6 | 96.5 |

tiplication; removing $\gamma$ makes SeeDNorm equivalent to directly replacing scaling factor of RMSNorm with the self-rescaling matrix.

**Impact of Multihead SeeDNorm.** In vision tasks, we employ multi-head SeeDNorm to reduce gradient variance and enhance training stability. In Table 4, we experiment with varying the number of heads for the image classification task. When the number of head is 1, the model fails to converge. And the results indicate that increasing the number of heads contributes to improving performance,

but an excessively high number can lead to reduced gradient diversity, thereby degrading performance. On larger models with greater hidden dimensions, SeeDNorm can similarly utilize a higher number of heads.

**Whether to apply weight decay to $\alpha$ and $\beta$.** Our previous analysis suggests that regularizing $\alpha$ and $\beta$ enhances gradient stability. Thus, we test omitting weight decay for $\alpha$ and $\beta$ in supervised image classification task. Results in Table 4 indicate that removing weight decay of $\alpha$ and $\beta$ will lead to inferior performance. And the result validates our theoretical analysis.

**Applying SeeDNorm in Each Attention Head.** When applying SeeDNorm in QKNorm in the task of language modeling, we perform normalization on each attention head, computing it in the same dimension as multi-head attention. In Table 3, we experiment with retaining the original structure of OLMoE, performing normalization across the entire hidden dimension. The results indicate that normalization on each attention head yields slightly better performance, though the difference is not significant.

**Multihead SeeDNorm in OLMoE.** In OLMoE, we do not use multi-head SeeDNorm. On one hand, training on a large corpus is less prone to overfitting and does not exhibit the high gradient variance seen in vision tasks with multiple training epochs. On the other hand, MoE models require appropriate gradient variance to dynamically train more experts. In Table 3, we also conducted experiments using a 16-head SeeDNorm configuration. The application of multihead SeeDNorm does not improve the performance.

**Combine with Advanced Structure.** In Table 3, we also presents performance evaluations of SeeDNorm on more advanced model architectures. For these experiments, we selected OLMoE variants improved with Frac-Connection (Zhu et al., 2025a). We conduct experiments on models with both 1.3B and 7B parameters, where the frac-rate is set to 2 for the 1.3B model and 4 for the 7B model, respectively. SeeDNorm can further enhance performance, with the effect being more pronounced in downstream tasks.

## 5    CONCLUSION

In this paper, we propose a novel normalization method SeeDNorm. SeeDNorm dynamically adjusts the scaling factor based on the input as a condition, thereby incorporating input norm information during the forward pass, which is overlooked in previous normalization layers like RMSNorm, while also enhancing the model's adaptability to diverse inputs. During backpropagation, SeeDNorm retains the capability to dynamically adjust gradients based on input magnitude. Experimental results demonstrate that SeeDNorm achieves faster convergence and superior performance compared to previously commonly used normalization layers or saturated activation functions across various tasks in language modeling and computer vision. We hope that this work will draw more attention to try to improve current normalization layers.

**Limitations and Future Work.** To demonstrate the intrinsic effectiveness of SeeDNorm and its robustness to hyperparameters, we use the same configuration for all normalization layers across all models, except that multi-head variants use different numbers of heads for different model sizes. However, in modern large language models, the optimal initialization strategy and other applicable techniques often vary across different normalization layers, which remains to be further explored. Furthermore, this paper verifies that improving normalization layers can bring substantial performance gains, but we introduce only a very small number of additional parameters. Therefore, we believe that investigating how to effectively scale up the parameter capacity of normalization layers to further improve the performance is a highly promising research direction.

## REPRODUCIBILITY STATEMENT

In Section 3, we present a detailed, equation-level specification of the proposed method, and we provide PyTorch implementation pseudocode in Algorithm 1 and Algorithm 2, as well as Triton implementation pseudocode in Algorithm 3 to facilitate faithful reproduction. The Appendix D further details the configuration of each experiment and all optimization hyperparameters, thereby ensuring result reproducibility. We also include comprehensive loss curves on various validation sets and accuracy curves on downstream tasks for side-by-side comparison.

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

APPENDIX

## A  THE USE OF LARGE LANGUAGE MODELS (LLMs)

During the writing process, we use LLMs only to assist in checking English spelling and grammar, as well as to standardize academic writing.

## B  GRADIENT RELATIONSHIP BETWEEN RMSNORM AND DyT

Notably, inspired by Stollenwerk (2025), dynamic $\tanh$ (Zhu et al., 2025b) and RMSNorm exhibit profound theoretical connections with regrad to the gradient in backpropagation.

**Proposition B.1.** *In backpropagation, DyT is an approximate element-wise operation of RMSNorm under the assumption that the norm of the input vector is constant.*

**Proof.**  Let $r = \frac{x}{\text{RMS}(x)}$, where $x \in \mathbb{R}^{1 \times D}$; it has been proven in (Zhang & Sennrich, 2019) that $\nabla_x r = \frac{I}{\text{RMS}(x)} - \frac{x^T x}{D \cdot \text{RMS}^3(x)}$. Since $\text{RMS}(x) = \frac{1}{\sqrt{D}} ||x||$, we conduct the derivation from the perspective of gradient equivalence:

$$\nabla_x r = \frac{\sqrt{D}}{||x||} \left( I - \frac{x^T x}{||x||^2} \right) = \frac{1}{\text{RMS}(x)} \left( I - \frac{(\sqrt{D} x^T)(\sqrt{D} x)}{D ||x||^2} \right) = \frac{1}{\text{RMS}(x)} \left( I - \frac{r^T r}{D} \right) \quad (9)$$

Given $\text{RMS}(x)$ that is a constant, let it be denoted as $c$. The operation $r_d = \frac{x_d}{\text{RMS}(x)}$ at each position can be treated as an independent computation, and Equation 9 can be written as an element-wise differential equation as follow:

$$\frac{dr_d}{dx_d} = \frac{1}{c} \left( 1 - \frac{r_d^2}{D} \right) \quad (10)$$

The steps to solve this differential equation are as follows:

$$\frac{dr_d}{dx_d} = \frac{1}{c} \left( 1 - \frac{r_d^2}{D} \right) \Rightarrow \frac{D}{D - r_d^2} dr_d = \frac{1}{c} dx_d \quad (11)$$

We integrate both sides of the equation, for notational convenience, we set all integration constants in the differential equation to zero by default:

$$\int_{r_d} \frac{D}{D - r_d^2} dr_d = \int_x \frac{1}{c} dx_d$$

$$D \cdot \frac{1}{2\sqrt{D}} \ln \left| \frac{\sqrt{D} + r_d}{\sqrt{D} - r_d} \right| = \frac{1}{c} x_d$$

$$\ln \left| \frac{\sqrt{D} + r_d}{\sqrt{D} - r_d} \right| = \frac{2 x_d}{c \sqrt{D}} \quad (12)$$

$$\left| \frac{\sqrt{D} + r_d}{\sqrt{D} - r_d} \right| = e^{\frac{2 x_d}{c \sqrt{D}}}$$

Since $-\sqrt{D} < r_d < \sqrt{D}$, the left side of the Equation 12 is necessarily greater than 0, the absolute value symbol can be removed. Then we have:

$$\frac{\sqrt{D} + r_d}{\sqrt{D} - r_d} = e^{\frac{2 x_d}{c \sqrt{D}}}$$

$$\Rightarrow r_d = \sqrt{D} \cdot \frac{e^{\frac{2 x_d}{c \sqrt{D}}} - 1}{e^{\frac{2 x_d}{c \sqrt{D}}} + 1} \quad (13)$$

Since $\tanh(z) = \frac{e^z - e^{-z}}{e^z + e^{-z}}$, we have:

$$
\begin{aligned}
r_d &= \sqrt{D} \cdot \frac{e^{\frac{2x_d}{c\sqrt{D}}} - 1}{e^{\frac{2x_d}{c\sqrt{D}}} + 1} \\
&= \sqrt{D} \cdot \frac{e^{\frac{x_d}{c\sqrt{D}}} - e^{-\frac{x_d}{c\sqrt{D}}}}{e^{\frac{x_d}{c\sqrt{D}}} + e^{-\frac{x_d}{c\sqrt{D}}}} \\
&= \sqrt{D} \cdot \tanh\left(\frac{x_d}{c\sqrt{D}}\right)
\end{aligned}
\tag{14}
$$

Since DyT includes a learnable scaling coefficient $\boldsymbol{\gamma}$, the constant $\sqrt{D}$ can be absorbed into $\boldsymbol{\gamma}$. Similarly, $\frac{1}{c\sqrt{D}}$ can also be incorporated into $\boldsymbol{\alpha}$.

Consequently, although DyT preserves the input norm in the forward pass, it loses the ability to dynamically adjust the gradient scale based on the magnitude of $\boldsymbol{x}$ during backpropagation, compared to RMSNorm. In contrast, our method retains norm information in both the forward and backward phases, endowing the model with data-dependent, self-rescaling gradients throughout the entire optimization.

## C  DETAILS OF GRADIENT ANALYSIS

### C.1  DETAILS OF GRADIENT ANALYSIS OF $\gamma$

Given the standard form of SeeDNorm as presented in Equation 15:

$$
\mathbf{SeeDNorm}(\boldsymbol{x}) = [\sigma(\boldsymbol{x} \cdot \boldsymbol{\beta}^T) \cdot \boldsymbol{\alpha} + \boldsymbol{\gamma}] \odot \frac{\boldsymbol{x}}{\mathrm{RMS}(\boldsymbol{x})}, \quad \text{where } \mathrm{RMS}(\boldsymbol{x}) = \sqrt{\frac{1}{D}\sum_{i=1}^{D} x_i^2}
\tag{15}
$$

we primarily investigate its gradient with respect to each token $\boldsymbol{x} \in \mathbb{R}^{1 \times D}$. For the input sequence $\boldsymbol{X} = [\boldsymbol{x}_1, \boldsymbol{x}_2, ..., \boldsymbol{x}_N] \in \mathbb{R}^{N \times D}$, since the computation of SeeDNorm for each token $\boldsymbol{x}_i$ does not interfere with others, the gradient calculation can be performed by simply concatenating the results computed for each token.

The gradient of the SeeDNorm output with respect to $\boldsymbol{\gamma}$ can be expressed as:

$$
\begin{aligned}
\frac{\partial\,\mathbf{SeeDNorm}(\boldsymbol{x})}{\partial\,\boldsymbol{\gamma}} &= \frac{\partial\left([\sigma(\boldsymbol{x} \cdot \boldsymbol{\beta}^T) \cdot \boldsymbol{\alpha}] \odot \frac{\boldsymbol{x}}{\mathrm{RMS}(\boldsymbol{x})}\right)}{\partial\,\boldsymbol{\gamma}} + \frac{\partial\left(\boldsymbol{\gamma} \odot \frac{\boldsymbol{x}}{\mathrm{RMS}(\boldsymbol{x})}\right)}{\partial\,\boldsymbol{\gamma}} \\
&= \mathrm{diag}\left(\frac{\boldsymbol{x}}{\mathrm{RMS}(\boldsymbol{x})}\right)
\end{aligned}
\tag{16}
$$

Here, $\frac{\boldsymbol{x}}{\mathrm{RMS}((\boldsymbol{x}))} \in \mathbb{R}^{1 \times D}$, and diag refers to the generation of a $D \times D$ diagonal matrix, where the diagonal elements $(i, i)$ correspond to the $i$-th element of $\frac{\boldsymbol{x}}{\mathrm{RMS}((\boldsymbol{x}))}$. During the actual backpropagation update of $\boldsymbol{\gamma}$, assuming the overall loss of the network is $\boldsymbol{L}$, the gradient of $\boldsymbol{\gamma}$ is given by:

$$
\nabla_{\boldsymbol{\gamma}} L = \frac{\partial\,\boldsymbol{L}}{\partial\,\mathbf{SeeDNorm}(\boldsymbol{x})} \cdot \frac{\partial\,\mathbf{SeeDNorm}(\boldsymbol{x})}{\partial\,\boldsymbol{\gamma}}
\tag{17}
$$

where $\frac{\partial\,\boldsymbol{L}}{\partial\,\mathbf{SeeDNorm}(\boldsymbol{x})} \in \mathbb{R}^{1 \times D}$, $\frac{\partial\,\mathbf{SeeDNorm}(\boldsymbol{x})}{\partial\,\boldsymbol{\gamma}} \in \mathbb{R}^{D \times D}$, and $\cdot$ denotes matrix multiplication. The final result $\nabla_{\boldsymbol{\gamma}} \boldsymbol{L} \in \mathbb{R}^{1 \times D}$ is the update tensor for $\boldsymbol{\gamma}$.

**Gradient of Multihead SeeDNorm.** When employing the multi-head variant of SeeDNorm, since $\boldsymbol{\gamma}$ does not participate in the per-head computation of the dynamic component $\sigma(\boldsymbol{x} \cdot \boldsymbol{\beta}^T) \cdot \boldsymbol{\alpha}$, the gradient with respect to $\boldsymbol{\gamma}$ remains identical to that of the standard (single-head) SeeDNorm.

## C.2 DETAILS OF GRADIENT ANALYSIS OF $\alpha$

For simplicity, we denote $\boldsymbol{F} = \textbf{SeeDNorm}(\boldsymbol{x}) \in \mathbb{R}^{1 \times D}$, $\boldsymbol{s} = [\sigma(\boldsymbol{x} \cdot \boldsymbol{\beta}^T) \cdot \boldsymbol{\alpha}] \in \mathbb{R}^{1 \times D}$ and $\boldsymbol{r} = \frac{\boldsymbol{x}}{\text{RMS}(\boldsymbol{x})} \in \mathbb{R}^{1 \times D}$. The gradient of the SeeDNorm output with respect to $\boldsymbol{\alpha}$ can be expressed as:

$$
\begin{aligned}
\frac{\partial\, \textbf{SeeDNorm}(\boldsymbol{x})}{\partial\, \boldsymbol{\alpha}} = \frac{\partial\, \boldsymbol{F}}{\partial\, \boldsymbol{\alpha}} &\triangleq \left( \frac{\partial F_k}{\partial \alpha_l} \right)_{k,l=1..D} \\
&= \frac{\partial}{\partial \alpha_l} \left[ (s_k + \gamma_k) r_k \right]_{k,l=1..D} \\
&= \left( r_k \cdot \frac{\partial s_k}{\partial \alpha_l} + r_k \cdot \frac{\partial \gamma_k}{\partial \alpha_l} \right)_{k,l=1..D} \\
&= \left( r_k \cdot \sigma(\boldsymbol{x} \cdot \boldsymbol{\beta}^T) \cdot \frac{\partial \alpha_k}{\partial \alpha_l} + 0 \right)_{k,l=1..D} \\
&= \left( \sigma(\boldsymbol{x} \cdot \boldsymbol{\beta}^T) r_k \cdot \delta_{kl} \right)_{k,l=1..D} \\
&= \boldsymbol{r} \cdot \left[ \sigma(\boldsymbol{x} \cdot \boldsymbol{\beta}^T) \boldsymbol{I}_{D \times D} \right] \\
&= \frac{\boldsymbol{x}}{\text{RMS}(\boldsymbol{x})} \cdot \left[ \sigma(\boldsymbol{x} \cdot \boldsymbol{\beta}^T) \boldsymbol{I}_{D \times D} \right]
\end{aligned}
\tag{18}
$$

Here, $\delta_{kl}$ is the Kronecker delta function, which equals 1 when $k = l$ and 0 otherwise; $\boldsymbol{I}_{D \times D}$ represents the $D \times D$ identity matrix, $F_k$, $s_k$, and $r_k$ represent the $k$-th elements of $\boldsymbol{F}$, $\boldsymbol{s}$, and $\boldsymbol{r}$, respectively, while $\alpha_l$ denotes the $l$-th element of $\boldsymbol{\alpha}$. Similar to the gradient of $\boldsymbol{\gamma}$, the final gradient of the SeeDNorm output with respect to $\boldsymbol{\alpha}$ is $\frac{\partial\, \textbf{SeeDNorm}(\boldsymbol{x})}{\partial\, \boldsymbol{\alpha}} \in \mathbb{R}^{D \times D}$, and during backpropagation, $\nabla_{\boldsymbol{\alpha}} \boldsymbol{L} = [\frac{\partial\, \boldsymbol{L}}{\partial\, \textbf{SeeDNorm}(\boldsymbol{x})} \cdot \frac{\partial\, \textbf{SeeDNorm}(\boldsymbol{x})}{\partial\, \boldsymbol{\alpha}}] \in \mathbb{R}^{1 \times D}$.

**Gradient of Multihead SeeDNorm.** When adopting the multi-head formulation, the derivation in Equation 18 begins to differ from $\frac{\partial s_k}{\partial \alpha_l}$ in the third line onward. We define the number of split heads as $n$, with $\boldsymbol{x} = [\boldsymbol{x}_{h_1}, ..., \boldsymbol{x}_{h_n}]$, where $\boldsymbol{x}_{h_i} \in \mathbb{R}^{1 \times \frac{D}{n}}$. Similarly, $\boldsymbol{\alpha} = [\boldsymbol{\alpha}_{h_1}, ..., \boldsymbol{\alpha}_{h_n}]$, $\boldsymbol{\beta} = [\boldsymbol{\beta}_{h_1}, ..., \boldsymbol{\beta}_{h_n}]$. Assuming $k$-th element and $l$-th element belong to the $i$-th head and $j$-th head, respectively, when $i \neq j$, $\frac{\partial s_k}{\partial \alpha_l} = 0$; when $i = j$, $\frac{\partial s_k}{\partial \alpha_l} = \sigma(\boldsymbol{\alpha}_{h_i} \cdot \boldsymbol{\beta}_{h_j}^T)$. The derivation is as follows:

$$
\begin{aligned}
\left( r_k \cdot \frac{\partial s_k}{\partial \alpha_l} + r_k \cdot \frac{\partial \gamma_k}{\partial \alpha_l} \right)_{k,l=1..D} &= \left( r_k \cdot \delta_{ij} \sigma(\boldsymbol{x}_{h_i} \cdot \boldsymbol{\beta}_{h_j}^T) \cdot \frac{\partial \alpha_k}{\partial \alpha_l} + 0 \right)_{k,l=1..D} \\
&= \left( \delta_{ij} \sigma(\boldsymbol{x}_{h_i} \cdot \boldsymbol{\beta}_{h_j}^T) r_k \cdot \delta_{kl} \right)_{k,l=1..D} \\
&= \left( \sigma(\boldsymbol{x}_{h_i} \cdot \boldsymbol{\beta}_{h_j}^T) r_k \cdot \delta_{kl} \right)_{k,l=1..D} \\
&= \boldsymbol{r} \cdot \begin{bmatrix} \sigma(\boldsymbol{x}_{h_1} \cdot \boldsymbol{\beta}_{h_1}^T) \boldsymbol{I}_{\frac{D}{n} \times \frac{D}{n}} & & \\ & \ddots & \\ & & \sigma(\boldsymbol{x}_{h_n} \cdot \boldsymbol{\beta}_{h_n}^T) \boldsymbol{I}_{\frac{D}{n} \times \frac{D}{n}} \end{bmatrix} \\
&= \frac{\boldsymbol{x}}{\text{RMS}(\boldsymbol{x})} \cdot \begin{bmatrix} \sigma(\boldsymbol{x}_{h_1} \cdot \boldsymbol{\beta}_{h_1}^T) \boldsymbol{I}_{\frac{D}{n} \times \frac{D}{n}} & & \\ & \ddots & \\ & & \sigma(\boldsymbol{x}_{h_n} \cdot \boldsymbol{\beta}_{h_n}^T) \boldsymbol{I}_{\frac{D}{n} \times \frac{D}{n}} \end{bmatrix}
\end{aligned}
\tag{19}
$$

## C.3 DETAILS OF GRADIENT ANALYSIS OF $\beta$

The gradient of the SeeDNorm output with respect to $\boldsymbol{\beta}$ can be expressed as:

$$
\begin{aligned}
\frac{\partial \, \mathbf{SeeDNorm}(\boldsymbol{x})}{\partial \, \boldsymbol{\beta}} &= \frac{\partial \, \boldsymbol{F}}{\partial \, \boldsymbol{\beta}} \triangleq \left( \frac{\partial F_k}{\partial \beta_l} \right)_{k,l=1..D} \\
&= \frac{\partial}{\partial \beta_l} \left[ (s_k + \gamma_k) \, r_k \right]_{k,l=1..D} \\
&= \left( r_k \cdot \frac{\partial s_k}{\partial \beta_l} + r_k \cdot \frac{\partial \gamma_k}{\partial \beta_l} \right)_{k,l=1..D} \\
&= \left( \alpha_k r_k \cdot \frac{\partial \sigma(\boldsymbol{x} \cdot \boldsymbol{\beta}^T)}{\partial \beta_l} + 0 \right)_{k,l=1..D} \\
&= \left[ \alpha_k r_k \cdot \frac{\partial}{\partial \beta_l} \sigma \left( \sum_{t=1}^{D} x_t \beta_t \right) \right]_{k,l=1..D} \\
&= \left[ \alpha_k r_k \cdot \sigma'(\boldsymbol{x} \cdot \boldsymbol{\beta}^T) \cdot \frac{\partial}{\partial \beta_l} \left( \sum_{t=1}^{D} x_t \beta_t \right) \right]_{k,l=1..D} \\
&= \left( \alpha_k r_k \cdot \sigma'(\boldsymbol{x} \cdot \boldsymbol{\beta}^T) \cdot x_l \right)_{k,l=1..D} \\
&= \sigma'(\boldsymbol{x} \cdot \boldsymbol{\beta}^T) \left( (\boldsymbol{\alpha} \odot \boldsymbol{r})^T \cdot \boldsymbol{x} \right) \\
&= \sigma'(\boldsymbol{x} \cdot \boldsymbol{\beta}^T) \left( \left( \boldsymbol{\alpha} \odot \frac{\boldsymbol{x}}{\mathrm{RMS}(\boldsymbol{x})} \right)^T \cdot \boldsymbol{x} \right)
\end{aligned}
\tag{20}
$$

where $\sigma'(\cdot)$ denotes the derivative of $\sigma(\cdot)$, when $\sigma(\cdot)$ is $\tanh$, the above expression can also be written as:

$$
\frac{\partial \, \mathbf{SeeDNorm}(\boldsymbol{x})}{\partial \, \boldsymbol{\beta}} = (1 - \tanh^2(\boldsymbol{x} \cdot \boldsymbol{\beta}^T)) \left( \left( \boldsymbol{\alpha} \odot \frac{\boldsymbol{x}}{\mathrm{RMS}(\boldsymbol{x})} \right)^T \cdot \boldsymbol{x} \right)
\tag{21}
$$

Similar to the gradients with respect to $\boldsymbol{\alpha}$ and $\boldsymbol{\gamma}$, the gradient of the SeeDNorm output with respect to $\boldsymbol{\beta}$ is also a $D \times D$ matrix, and the final gradient for updating $\boldsymbol{\beta}$ in backpropagation is given by $\nabla_{\boldsymbol{\beta}} \boldsymbol{L} = [\frac{\partial \, \boldsymbol{L}}{\partial \, \mathbf{SeeDNorm}(\boldsymbol{x})} \cdot \frac{\partial \, \mathbf{SeeDNorm}(\boldsymbol{x})}{\partial \, \boldsymbol{\beta}}] \in \mathbb{R}^{1 \times D}$.

**Gradient of Multihead SeeDNorm.** Similar to the derivation for $\boldsymbol{\alpha}$, the main difference in the gradient of $\boldsymbol{\beta}$ under the multi-head form also lies in $\frac{\partial s_k}{\partial \beta_l}$ in the third line of Equation 20. We also define that the $k$-th element and the $l$-th element belong to the $i$-th and $j$-th heads, respectively. The derivation under the multi-head form is as follows:

$$
\begin{aligned}
\left( r_k \cdot \frac{\partial s_k}{\partial \beta_l} + r_k \cdot \frac{\partial \gamma_k}{\partial \beta_l} \right)_{k,l=1..D} &= \left( \alpha_k r_k \delta_{ij} \cdot \frac{\partial \sigma(\boldsymbol{x}_{h_i} \cdot \boldsymbol{\beta}_{h_j}^T)}{\partial \beta_l} + 0 \right)_{k,l=1..D} \\
&= \left[ \alpha_k r_k \delta_{ij} \cdot \frac{\partial}{\partial \beta_l} \sigma \left( \sum_{t=1}^{\frac{D}{n}} x_{h_i,t} \beta_{h_j,t} \right) \right]_{k,l=1..D} \\
&= \left[ \alpha_k r_k \delta_{ij} \cdot \sigma'(\boldsymbol{x}_{h_i} \cdot \boldsymbol{\beta}_{h_j}^T) \cdot \frac{\partial}{\partial \beta_l} \sigma \left( \sum_{t=1}^{\frac{D}{n}} x_{h_i,t} \beta_{h_j,t} \right) \right]_{k,l=1..D} \\
&= \left( \alpha_k r_k \delta_{ij} \cdot \sigma'(\boldsymbol{x}_{h_i} \cdot \boldsymbol{\beta}_{h_j}^T) \cdot \delta_{ij} x_l \right)_{k,l=1..D} \\
&= \left( \alpha_k r_k \cdot \delta_{ij} \sigma'(\boldsymbol{x}_{h_i} \cdot \boldsymbol{\beta}_{h_j}^T) \cdot x_l \right)_{k,l=1..D} \\
&= \left( \boldsymbol{\alpha} \odot \frac{\boldsymbol{x}}{\mathrm{RMS}(\boldsymbol{x})} \right)^T \cdot \left[ \sigma'(\boldsymbol{x}_{h_1} \cdot \boldsymbol{\beta}_{h_1}^T) \boldsymbol{x}_{h_1} \quad \ldots \quad \sigma'(\boldsymbol{x}_{h_n} \cdot \boldsymbol{\beta}_{h_n}^T) \boldsymbol{x}_{h_n} \right]
\end{aligned}
\tag{22}
$$

## C.4 Details of gradient analysis of $x$

Beyond the update of learnable parameters in SeeDNorm, in this section, we also analyze the impact of SeeDNorm as a component in the overall backpropagation of the network by deriving the gradient of the SeeDNorm with respect to its input $x$. The gradient of the SeeDNorm output with respect to $x$ can be expressed as:

$$
\begin{aligned}
\frac{\partial \, \mathbf{SeeDNorm}(x)}{\partial \, x} = \frac{\partial \, F}{\partial \, x} &\triangleq \left( \frac{\partial F_k}{\partial x_l} \right)_{k,l=1..D} \\
&= \frac{\partial}{\partial x_l} \left[ (s_k + \gamma_k) \, r_k \right]_{k,l=1..D} \\
&= \left[ r_k \cdot \frac{\partial s_k}{\partial x_l} + (s_k + \gamma_k) \cdot \frac{\partial r_k}{\partial x_l} \right]_{k,l=1..D} \\
&= \left[ \alpha_k r_k \cdot \frac{\partial \sigma(x \cdot \boldsymbol{\beta}^T)}{\partial x_l} + (s_k + \gamma_k) \cdot \frac{\partial r_k}{\partial x_l} \right]_{k,l=1..D}
\end{aligned}
\tag{23}
$$

For the first term, we have the following derivation:

$$
\begin{aligned}
\frac{\partial \sigma(x \cdot \boldsymbol{\beta}^T)}{\partial x_l} &= \frac{\partial}{\partial x_l} \sigma \left( \sum_{t=1}^{D} x_t \beta_t \right) \\
&= \sigma'(x \cdot \boldsymbol{\beta}^T) \frac{\partial}{\partial x_l} \left( \sum_{t=1}^{D} x_t \beta_t \right) \\
&= \sigma'(x \cdot \boldsymbol{\beta}^T) \beta_l
\end{aligned}
\tag{24}
$$

For the second term, we have the following derivation:

$$
\begin{aligned}
\frac{\partial r_k}{\partial x_l} &= \frac{\partial}{\partial x_l} \left( \frac{x_k}{\mathrm{RMS}(x)} \right) \\
&= \frac{\delta_{kl} \cdot \mathrm{RMS}(x) - x_k \cdot \frac{x_l}{D \cdot \mathrm{RMS}(x)}}{\mathrm{RMS}^2(x)} \\
&= \frac{\delta_{kl}}{\mathrm{RMS}(x)} - \frac{x_k x_l}{D \cdot \mathrm{RMS}^3(x)}
\end{aligned}
\tag{25}
$$

By substituting Equations 24 and 25 into Equation 23, we obtain:

$$
\begin{aligned}
\frac{\partial \, \mathbf{SeeDNorm}(x)}{\partial \, x} &\triangleq \left[ \alpha_k r_k \sigma'(x \cdot \boldsymbol{\beta}^T) \beta_l + (s_k + \gamma_k) \left( \frac{\delta_{kl}}{\mathrm{RMS}(x)} - \frac{x_k x_l}{D \cdot \mathrm{RMS}^3(x)} \right) \right]_{k,l=1..D} \\
&= \sigma'(x \cdot \boldsymbol{\beta}^T)(\boldsymbol{\alpha} \odot \boldsymbol{r})^T \cdot \boldsymbol{\beta} + \frac{1}{\mathrm{RMS}(x)} \mathrm{diag}(\boldsymbol{s} + \boldsymbol{\gamma}) - \frac{(\boldsymbol{s}+\boldsymbol{\gamma})^T \mathbf{1}_{1 \times D}}{D \cdot \mathrm{RMS}^3(x)} \odot (\boldsymbol{x}^T \cdot \boldsymbol{x}) \\
&= \sigma'(x \cdot \boldsymbol{\beta}^T)(\boldsymbol{\alpha} \odot \frac{\boldsymbol{x}}{\mathrm{RMS}(x)})^T \cdot \boldsymbol{\beta} + \frac{1}{\mathrm{RMS}(x)} \mathrm{diag}(\boldsymbol{s} + \boldsymbol{\gamma}) - \frac{(\boldsymbol{s}+\boldsymbol{\gamma})^T \mathbf{1}_{1 \times D}}{D \cdot \mathrm{RMS}^3(x)} \odot (\boldsymbol{x}^T \cdot \boldsymbol{x})
\end{aligned}
\tag{26}
$$

**Gradient of Multihead SeeDNorm.** Consistent with the previous derivations, the main difference in the gradient of SeeDNorm with respect to $x$ in the multi-head form lies in the part corresponding to Equation 24. In the multi-head form, we have:

$$\frac{\partial s_k}{\partial x_l} = \delta_{ij} \frac{\partial \sigma(\boldsymbol{x}_{h_i} \cdot \boldsymbol{\beta}_{h_j}^T)}{\partial x_l}$$

$$= \delta_{ij} \frac{\partial}{\partial x_l} \sigma \left( \sum_{t=1}^{\frac{D}{n}} x_{h_i,t} \beta_{h_j,t} \right)$$

$$= \delta_{ij} \sigma'(\boldsymbol{x}_{h_i} \cdot \boldsymbol{\beta}_{h_j}^T) \frac{\partial}{\partial x_l} \left( \sum_{t=1}^{\frac{D}{n}} x_{h_i,t} \beta_{h_j,t} \right) \tag{27}$$

$$= \delta_{ij} \sigma'(\boldsymbol{x}_{h_i} \cdot \boldsymbol{\beta}_{h_j}^T) \beta_l$$

The final form of the gradient in the multi-head setting is:

$$(\boldsymbol{\alpha} \odot \frac{\boldsymbol{x}}{\text{RMS}(\boldsymbol{x})})^T \cdot \left[ \sigma'(\boldsymbol{x}_{h_1} \cdot \boldsymbol{\beta}_{h_1}^T) \boldsymbol{\beta}_{h_1} \quad \cdots \quad \sigma'(\boldsymbol{x}_{h_1} \cdot \boldsymbol{\beta}_{h_n}^T) \boldsymbol{\beta}_{h_n} \right] + \frac{1}{\text{RMS}(\boldsymbol{x})} \text{diag}(\boldsymbol{s} + \boldsymbol{\gamma}) - \frac{(\boldsymbol{s} + \boldsymbol{\gamma})^T \mathbf{1}_{1 \times D}}{D \cdot \text{RMS}^3(\boldsymbol{x})} \odot (\boldsymbol{x}^T \cdot \boldsymbol{x}) \tag{28}$$

## C.5 Discussion about Gradients

**Gradients of $\boldsymbol{\gamma}$.** From Equation 16, it can be observed that the gradient magnitude of SeeDNorm with respect to $\boldsymbol{\gamma}$ is not influenced by the scale of the input $\boldsymbol{x}$. Because $\frac{k\boldsymbol{x}}{\text{RMS}(k\boldsymbol{x})} = \frac{k\boldsymbol{x}}{\sqrt{\frac{1}{D} \sum_i^D (kx_i)^2}} = \frac{\boldsymbol{x}}{\text{RMS}(\boldsymbol{x})}$. Therefore, the gradient of $\boldsymbol{\gamma}$ exhibits scale invariance, remaining fundamentally stable without requiring additional processing.

**Gradients of $\boldsymbol{\alpha}$.** By contrast, as shown in Equation 18, the gradient of $\boldsymbol{\alpha}$ incorporates $\sigma(\boldsymbol{x} \cdot \boldsymbol{\beta}^T)$, which introduces scale-related information from $\boldsymbol{x}$ and consequently deprives the $\boldsymbol{\alpha}$ gradient of the scale invariance observed in $\boldsymbol{\gamma}$. Therefore, to ensure training stability, we need to constrain its range within a fixed interval using an activation function $\sigma$. Ultimately, we adopt the $\tanh$ function for this purpose. Additionally, $\boldsymbol{\alpha}$ directly multiplies with other terms in the gradients of both $\boldsymbol{\beta}$ and $\boldsymbol{x}$ without constraints. Therefore, we prefer the model to be more cautious when initiating updates for $\boldsymbol{\alpha}$. To achieve this, we initialize $\boldsymbol{\beta}$ to 0, ensuring that $\boldsymbol{\alpha}$ starts with a smaller gradient during the update process.

**Gradients of $\boldsymbol{\beta}$.** The gradient with respect to $\boldsymbol{\beta}$ further depends on $\boldsymbol{x}$ and $\boldsymbol{\alpha}$. When $\boldsymbol{x}$ is abnormally large, since $1 - \tanh^2(\boldsymbol{x})$ is a higher-order infinitesimal of $\frac{1}{\boldsymbol{x}}$, the gradient of $\boldsymbol{\beta}$ approaches 0 at this point. When $\boldsymbol{x}$ is abnormally small, the gradients of $\boldsymbol{\alpha}$ and $\boldsymbol{\beta}$ also approaches 0. This situation is rare in practice, and even if it occurs, $\boldsymbol{\gamma}$ can still be updated normally. And subsequent analysis also indicates that anomalous values of $\boldsymbol{x}$ do not have a catastrophic impact on the gradient of preceding layers during backpropagation when SeeDNorm is applied. Therefore, our primary concern is to prevent gradient explosion. Since $\boldsymbol{\alpha}$ directly affects the gradient of $\boldsymbol{\beta}$, we apply weight decay to $\boldsymbol{\alpha}$. Similarly, because $\boldsymbol{\beta}$ also influences the gradient of $\boldsymbol{x}$, we apply weight decay to $\boldsymbol{\beta}$ as well, to control their numerical stability.

**Gradients of the input $\boldsymbol{x}$.** Regarding the gradient of $\boldsymbol{x}$, the first term is similar to the gradient of $\boldsymbol{\beta}$, it incorporates information about the norm of $\boldsymbol{x}$ and uses $\sigma'$ to keep the values bounded. When $\boldsymbol{x}$ is abnormally large, $\sigma'(\boldsymbol{x}) = 1 - \tanh^2(\boldsymbol{x})$ approaches 0, and since $1 - \tanh^2(\boldsymbol{x})$ is a higher-order infinitesimal of $\frac{1}{\boldsymbol{x}}$, this ensures that the gradient does not explode. At this point, the gradient of SeeDNorm with respect to the input $\boldsymbol{x}$ is primarily dominated by the last two terms. Conversely, when $\boldsymbol{x}$ is abnormally small, $\sigma'(\boldsymbol{x})$ approaches 1, and numerical stability of this term can be maintained by constraining the values of $\boldsymbol{\alpha}$ and $\boldsymbol{\beta}$, and the gradient with respect to $\boldsymbol{x}$ is again dominated by the latter two terms.

For the last two terms of Equation 26, they can be expressed as $\frac{1}{\text{RMS}(\boldsymbol{x})} \left( \text{diag}(\boldsymbol{s} + \boldsymbol{\gamma}) - \frac{(\boldsymbol{s}+\boldsymbol{\gamma})^T \mathbf{1}_{1 \times D}}{D \cdot \text{RMS}^2(\boldsymbol{x})} \odot (\boldsymbol{x}^T \cdot \boldsymbol{x}) \right)$. When the sample $\boldsymbol{x}$ undergoes scaling, assuming $\boldsymbol{x}' = k\boldsymbol{x}$, the expression becomes:

$$\frac{1}{\text{RMS}(k\boldsymbol{x})}\left(\text{diag}(\sigma(k\boldsymbol{x}\cdot\boldsymbol{\beta}^T)\cdot\boldsymbol{\alpha}+\boldsymbol{\gamma})-\frac{(\sigma(k\boldsymbol{x}\cdot\boldsymbol{\beta}^T)\cdot\boldsymbol{\alpha}+\boldsymbol{\gamma})^T\mathbf{1}_{1\times D}}{D\cdot\text{RMS}^2(k\boldsymbol{x})}\odot(k\boldsymbol{x}^T\cdot k\boldsymbol{x})\right)$$

$$=\frac{1}{k\text{RMS}(\boldsymbol{x})}\left(\text{diag}(\sigma(k\boldsymbol{x}\cdot\boldsymbol{\beta}^T)\cdot\boldsymbol{\alpha}+\boldsymbol{\gamma})-\frac{k^2(\sigma(k\boldsymbol{x}\cdot\boldsymbol{\beta}^T)\cdot\boldsymbol{\alpha}+\boldsymbol{\gamma})^T\mathbf{1}_{1\times D}}{k^2 D\cdot\text{RMS}^2(\boldsymbol{x})}\odot(\boldsymbol{x}^T\cdot\boldsymbol{x})\right) \quad (29)$$

$$=\frac{1}{k\text{RMS}(\boldsymbol{x})}\left(\text{diag}(\sigma(k\boldsymbol{x}\cdot\boldsymbol{\beta}^T)\cdot\boldsymbol{\alpha}+\boldsymbol{\gamma})-\frac{(\sigma(k\boldsymbol{x}\cdot\boldsymbol{\beta}^T)\cdot\boldsymbol{\alpha}+\boldsymbol{\gamma})^T\mathbf{1}_{1\times D}}{D\cdot\text{RMS}^2(\boldsymbol{x})}\odot(\boldsymbol{x}^T\cdot\boldsymbol{x})\right)$$

When $k$ is abnormally large, $\sigma(\cdot)$ approaches 1 when $\sigma$ is implemented with `tanh`, and the numerator and denominator of $\frac{\boldsymbol{x}^T\cdot\boldsymbol{x}}{\text{RMS}^2(\boldsymbol{x})}$ are of the same order of magnitude. Therefore, the above expression is primarily influenced by $\frac{1}{k\text{RMS}(\boldsymbol{x})}$, and it scales down by a factor of $k$. Therefore, this achieves a form of adaptive stability. When the output scale of the previous layer $\boldsymbol{x}$ is abnormally large, the corresponding gradient in backpropagation decreases, thereby ensuring training stability.

When $k$ is abnormally small, $\tanh(k\boldsymbol{x}\cdot\boldsymbol{\beta}^T)$ and $k\cdot\text{RMS}(\boldsymbol{x})$ are equivalent infinitesimals. Therefore, the above expression is primarily influenced by $\frac{\boldsymbol{\gamma}}{k\text{RMS}(\boldsymbol{x})}$. As $k$ decreases, $\frac{\boldsymbol{\gamma}}{k\text{RMS}(\boldsymbol{x})}$ increases proportionally, thus also achieving adaptive gradient stability.

Table 5: All validation datasets and downstream test datasets used in OLMoE and OLMo2. For the validation set, our primary metrics are validation loss and perplexity (PPL). For downstream tasks, we conduct zero-shot evaluation and report answer accuracy (Acc%) as the key metric.

| Validation Datasets |
|---|
| `c4_en-validation` (Raffel et al., 2020) |
| `dolma_books-validation` (Soldaini et al., 2024) |
| `dolma_common-crawl-validation` (Soldaini et al., 2024) |
| `dolma_pes2o-validation` (Soldaini et al., 2024) |
| `dolma_reddit-validation` (Soldaini et al., 2024) |
| `dolma_stack-validation` (Soldaini et al., 2024) |
| `dolma_wiki-validation` (Soldaini et al., 2024) |
| `ice-validation` (Greenbaum & Nelson, 1996) |
| `m2d2_s2orc-validation` (Lo et al., 2020) |
| `pile-validation` (Gao et al., 2020) |
| `wiki_103-validation` (Merity et al., 2016) |
| **Downstream Tasks** |
| `PIQA` (Bisk et al., 2020) |
| `HellaSwag` (Zellers et al., 2019) |
| `ARC-Challenge` (Clark et al., 2018) |
| `ARC-Easy` (Clark et al., 2018) |
| `MMLU-Var` (Hendrycks et al.) |
| `Winogrande` (Sakaguchi et al., 2021) |
| `Openbook-QA` (Mihaylov et al., 2018) |
| `SCIQ` (Welbl et al., 2017) |
| `COPA` (Roemmele et al., 2011) |
| `BoolQ` (Clark et al., 2019) |
| `Commonsense_QA` (Talmor et al., 2019) |
| `Social_IQA` (Sap et al., 2019) |

## C.6 VARIANCE OF THE DOT-PRODUCT OF TWO RANDOM VECTORS

**Theorem 3.2.** *In high-dimensional space, the variance of the dot product of two random vectors is proportional to their dimension $D$.*

**Prof.** Suppose there are two $D$-dimensional random vectors $\boldsymbol{x} = [x_1, x_2, ..., x_D]$ and $\boldsymbol{y} = [y_1, y_2, ..., y_D]$. Their components are independent and identically distributed (i.i.d.) random variables, and they satisfy:

$$E(x_i) = E(y_i) = 0$$
$$\mathrm{Var}(x_i) = \mathrm{Var}(y_i) = \sigma^2 \tag{30}$$

Then $\mathrm{Var}(\boldsymbol{x} \cdot \boldsymbol{y}^T) = \mathrm{Var}(\sum_{i=1}^{D} x_i y_i)$, let $s = \boldsymbol{x} \cdot \boldsymbol{y}^T$, we have:

$$\mathrm{Var}(s) = E(s^2) - E^2(s) = E(\boldsymbol{s}^2)$$

$$E[s^2] = E\left[\left(\sum_{i=1}^{D} x_i y_i\right)^2\right] = E\left[\sum_{i=1}^{D}\sum_{j=1}^{D}(x_i y_i)(x_j y_j)\right] = \sum_{i=1}^{D}\sum_{j=1}^{D} E[x_i y_i x_j y_j] \tag{31}$$

$$\sum_{i=1}^{D}\sum_{j=1}^{D} E[x_i y_i x_j y_j] = \sum_{i=1}^{D}\sum_{j=1}^{D} \delta_{ij} E[x_i y_i x_j y_j] = DE[x_i^2 y_i^2] = DE[x_i^2]E[y_i^2]$$

Therefore, $E[s^2] = D\sigma^4$ is proportional to the dimension size.

## D  DETAILS OF EXPERIMENTS AND MODEL SETTINGS

In this section, we will provide more experimental results, a detailed description of the different model configurations and hyperparameter settings used for each task, as well as the parameter settings for SeeDNorm.

### D.1  OLMOE AND OLMO2 IN LANGUAGE MODELING

#### D.1.1  VALIDATION DATASETS AND DOWNSTREAM TASKS

The validation datasets and downstream task datasets used for OLMoE and OLMo2 in the language modeling task are presented in Table 5. For the validation datasets, we primarily focus on the validation loss and perplexity (PPL). Since the trends of PPL and loss are consistent, we mainly report the loss results. For the downstream task datasets, we report the answer accuracy rate of the model.

#### D.1.2  MODEL AND TRAINING SETTINGS

**OLMoE.** The model configurations and hyper-parameters used in our OLMoE-1.3B and OLMoE-7B models are presented in Table 6. The training and optimization parameters for both models are shown in Table 7.

Table 6: Configuration and hyperparameters for **OLMoE-1.3B** and **OLMoE-7B**.

| Model | OLMoE-1.3B | OLMoE-7B |
|---|---|---|
| Num Layer | 12 | 16 |
| Hidden Dim | 1024 | 2048 |
| Num Head | 16 | 16 |
| Position Embed | RoPE ($\theta = 10000$) | |
| Context Length | 4096 | 4096 |
| MoE Top-k | 8 | 8 |
| MoE Experts | 64 | 64 |
| Weight Tying | Yes | Yes |

Table 7: Hyperparameters for the optimizer of **OLMoE-1.3B** and **OLMoE-7B**.

| Model | OLMoE-1.3B | OLMoE-7B |
|---|---|---|
| Optimizer | AdamW ($\beta_1 = 0.9, \beta_2 = 0.95$) | |
| Learning Rate | 4.0e-4 | 4.0e-4 |
| Weight Decay | 0.1 | 0.1 |
| LR Schedule | Cosine | Cosine |
| Warm up Tokens | 10B | 10B |
| Balance Loss Weight | 0.01 | 0.01 |
| Router z-loss Weight | 0.001 | 0.001 |
| Gradient Clip | 1.0 | 1.0 |
| Micro BatchSize | 6 | 4 |
| Global BatchSize | 768 | 1024 |

OLMoE baseline employs a PreNorm-based structure, where normalization is applied at the input of both the attention and MoE layers. Additionally, there are query norm and key norm layers within the attention layer. A normalization layer is also present at the Transformer output. We replaced all these normalization layers from RMSNorm to our proposed SeeDNorm. Specifically, for QueryNorm and KeyNorm, we perform SeeDNorm in each attention head.

**OLMo2.** The model configurations and hyper-parameters used in our OLMo2-550M and OLMo2-1B models are presented in Table 8. The training and optimization parameters for both models are shown in Table 9. The application of the SeeDNorm is consistent with that in OLMoE.

Table 8: Configuration and hyperparameters for **OLMo2-550M** and **OLMo2-1B**.

| Model | OLMo2-550M | OLMo2-1B |
|---|---|---|
| Num Layer | 16 | 16 |
| Hidden Dim | 1536 | 2048 |
| Num Head | 16 | 32 |
| Num KV Head | 4 | 8 |
| Position Embed | RoPE ($\theta = 500000$) | |
| Context Length | 4096 | 4096 |
| Weight Tying | Yes | Yes |

Table 9: Hyperparameters for the optimizer of **OLMo2-550M** and **OLMo2-1B**.

| Model | OLMo2-550M | OLMo2-1B |
|---|---|---|
| Optimizer | AdamW ($\beta_1 = 0.9, \beta_2 = 0.95$) | |
| Learning Rate | 3.0e-4 | 4.0e-4 |
| Weight Decay | 0.1 | 0.1 |
| LR Schedule | Cosine | Cosine |
| Warm up Tokens | 8B | 8B |
| Gradient Clip | 1.0 | 1.0 |
| Micro BatchSize | 8 | 4 |
| Global BatchSize | 1024 | 1024 |

### D.1.3 EXPERIMENT RESULTS

**OLMoE.** Figure 5 summarizes the downstream-task comparison between OLMoE-1.3B equipped with SeeDNorm and the RMSNorm baseline. SeeDNorm yields consistent and often substantial gains across almost every task, delivering $> 2\times$ speed-up on multiple datasets such as ARC-Easy (Clark et al., 2018), ARC-Challenge (Clark et al., 2018), and Social-IQA (Sap et al., 2019). Figure 6 reports the analysis on all validation splits and shows equally pronounced advantages. Figure 7 summarizes the downstream-task comparison between OLMoE-7B equipped with SeeDNorm and the RMSNorm baseline. SeeDNorm also yields consistent and often substantial gains across almost every task in OLMoE-7B, delivering $> 2\times$ speed-up on multiple datasets such as ARC-Challenge (Clark et al., 2018), Social-IQA (Sap et al., 2019) and PIQA (Bisk et al., 2020). Figure 8 reports the analysis on all validation splits and shows equally pronounced advantages.

**OLMo2.** Figure 9 illustrates the comparison of validation loss between SeeDNorm and the baseline RMSNorm for OLMo2-550M across all validation sets. And Figure 10 illustrates the comparison of accuracy of downstream tasks between SeeDNorm and the baseline RMSNorm for OLMo2-1B across all validation sets. Our method similarly achieves consistent improvements in most validation datasets and downstream tasks.

By comparing the experimental results of OLMoE and OLMo2, we demonstrate that SeeDNorm achieves greater performance gains in MoE architectures. Given that MoE architectures have become the dominant paradigm for nearly all state-of-the-art LLMs developed in recent years, demonstrating robust performance in MoE settings is of paramount practical importance.

To elucidate why SeeDNorm is inherently better suited to MoE architectures than to dense models, we attribute this discrepancy to the dynamic nature of MoE routing: each token is processed by a distinct subset of experts, and the expert assignments vary significantly across different tokens, leading to real-time fluctuations in the input distribution of the normalization layer. The input-dependent scaling mechanism inherent to SeeDNorm is uniquely well-positioned to adapt to such distributional variations. In contrast, dense models exhibit far more uniform token distributions, which inherently limits the benefits conferred by this dynamic adaptation mechanism. We believe that exploring the application of SeeDNorm to other MoE-based models in various tasks (Cai et al., 2025; Shi et al., 2025; Xu et al., 2024; Yuan et al., 2025) represents a worthwhile research direction.

### D.2 VIT AND CONVNEXT IN IMAGE CLASSIFICATION

In the image classification task, we conduct training on the ImageNet-1K (Krizhevsky et al., 2012) training set and performed evaluation on the test set. The model configurations and hyper-parameters in our used ViT-B and ViT-L models in image classification task are presented in Table 10. The training and optimization parameters for both models are shown in Table 11.

In all ViT classification experiments, we employ the multi-head variant of SeeDNorm to enhance training stability, because the classification task requires hundreds of training epochs, models are prone to overfitting and can easily result in excessively high gradient variance (Pezeshki et al., 2021). When Multihead SeeDNorm is not used, larger models like ViT-L cannot even converge. To further

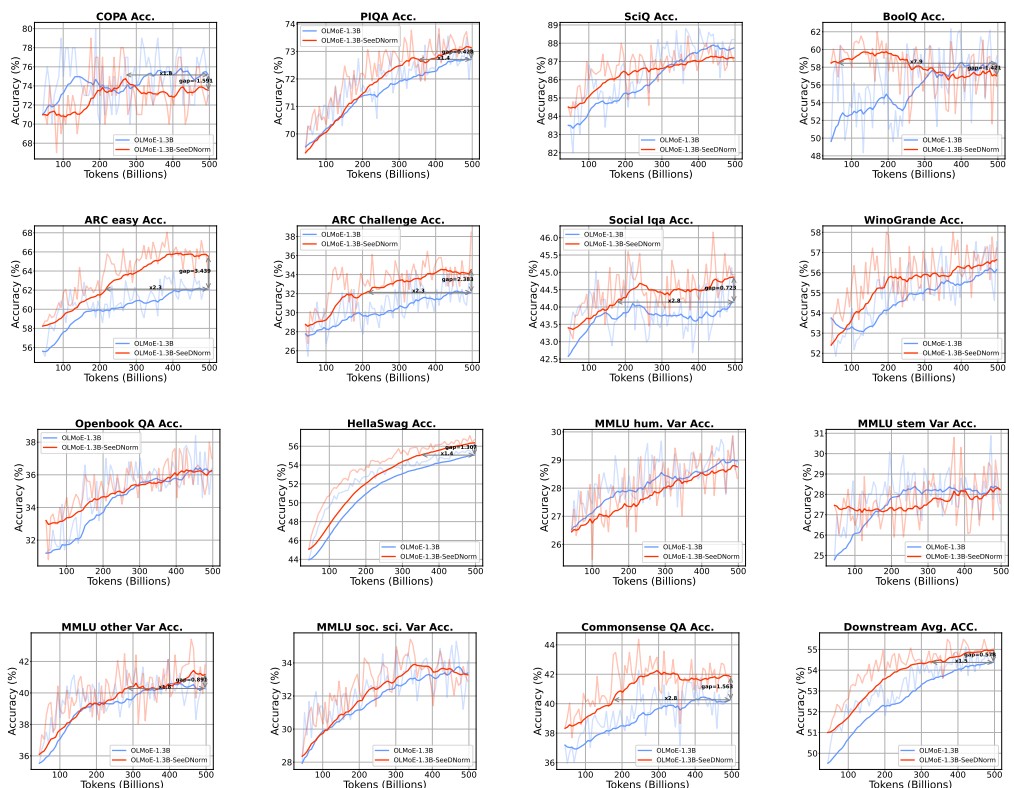

Figure 5: Comparisons of the accuracy of all downstream tasks from Table 5 in **OLMoE-1.3B** when using SeeDNorm as the normalization layer versus the default RMSNorm. The figure illustrates the evolution of downstream task accuracy as the total training tokens increase during training, with transparent lines indicating unsmoothed results and solid lines denoting 0.99 EMA-smoothed results.

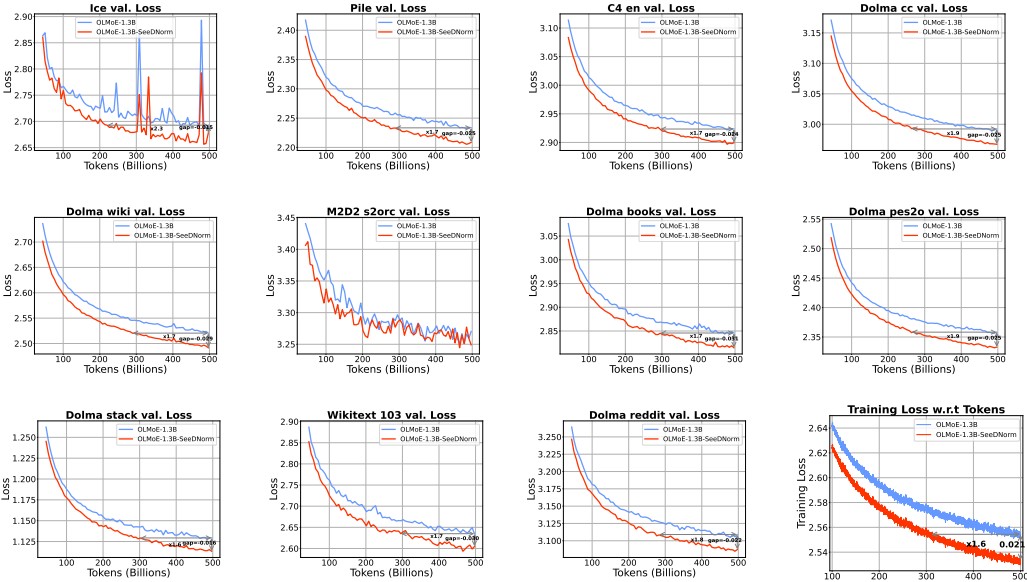

Figure 6: Comparisons of the validation CrossEntropy loss of all validation datasets from Table 5 in **OLMoE-1.3B** when using SeeDNorm as the normalization layer versus the default RMSNorm. The figure illustrates the evolution of the validation loss as the total training tokens increase during training.

stabilize the training, we additionally divide $\boldsymbol{\alpha} \cdot \boldsymbol{\beta}^T$ by the dimension and apply dropout to the whole dynamic coefficient $\sigma(\boldsymbol{x} \cdot \boldsymbol{\beta}^T) \cdot \boldsymbol{\alpha}$ of SeeDNorm; the dropout rate of SeeDNorm is set equal to the

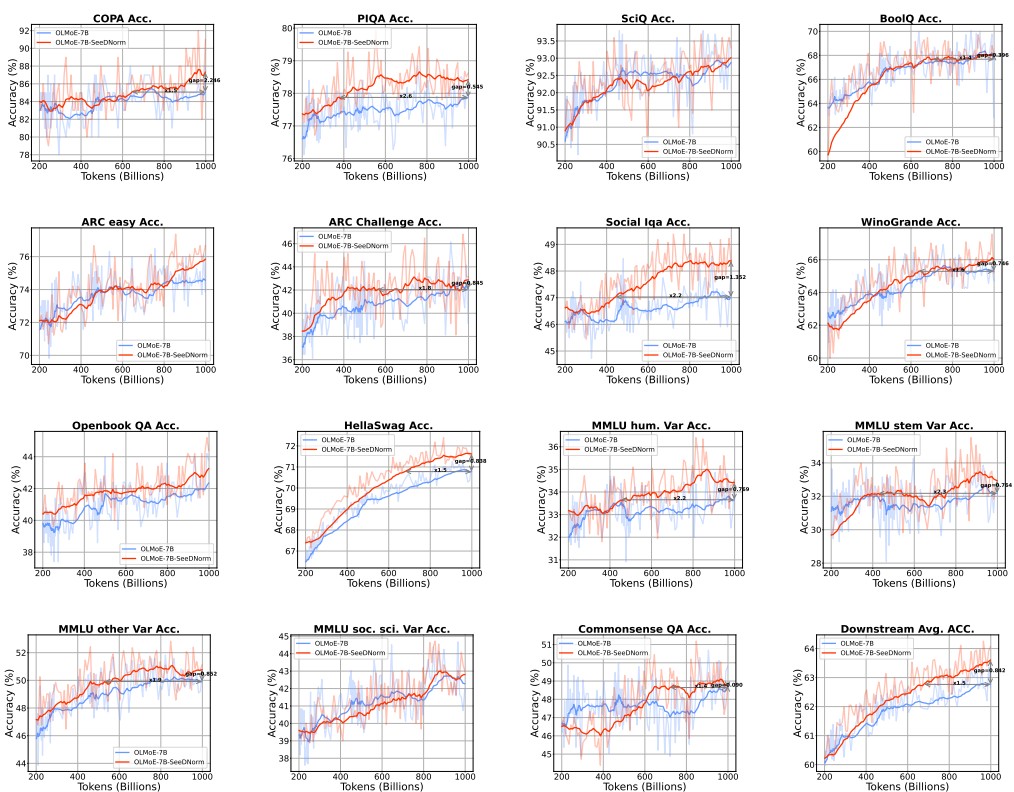

Figure 7: Comparisons of the accuracy of all downstream tasks from Table 5 in **OLMoE-7B** when using SeeDNorm as the normalization layer versus the default RMSNorm. The figure illustrates the evolution of downstream task accuracy as the total training tokens increase during training, with transparent lines indicating unsmoothed results and solid lines denoting 0.99 EMA-smoothed results.

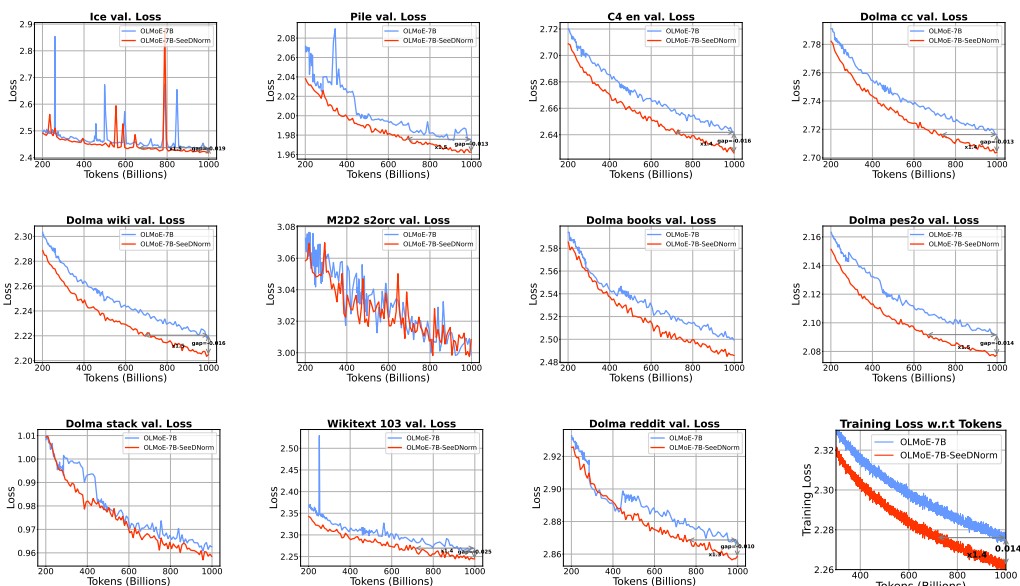

Figure 8: Comparisons of the validation CrossEntropy loss of all validation datasets from Table 5 in **OLMoE-7B** when using SeeDNorm as the normalization layer versus the default RMSNorm. The figure illustrates the evolution of the validation loss as the total training tokens increase during training.

drop-path rate of the model. Notably, ViT-L, when augmented with SeeDNorm, exhibits strong fitting

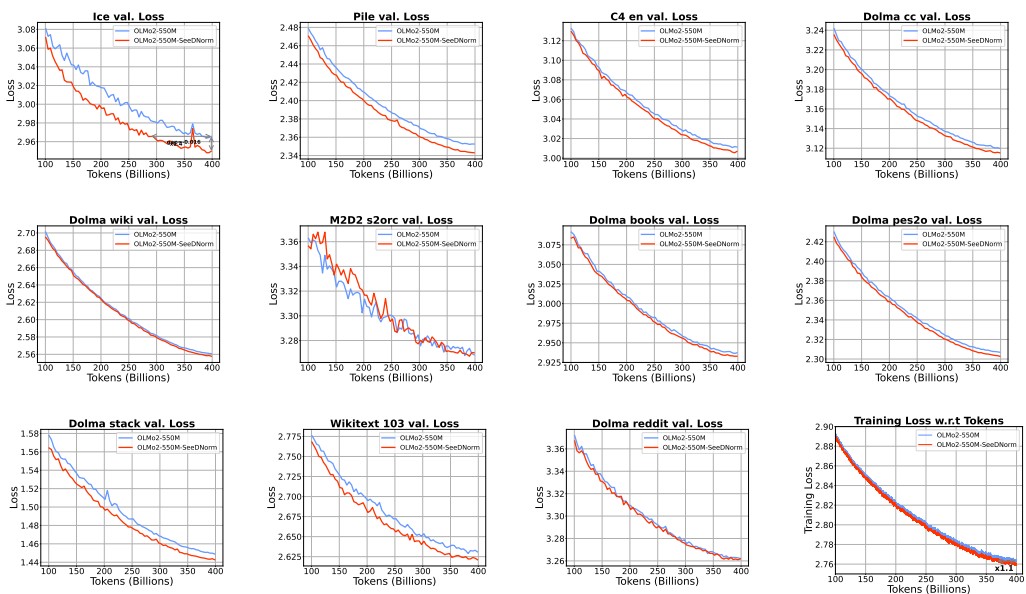

Figure 9: Comparisons of the validation CrossEntropy loss of all validation datasets from Table 5 in **OLMo2-550M** when using SeeDNorm as the normalization layer versus the default RMSNorm. The figure illustrates the evolution of the validation loss as the total training tokens increase during training.

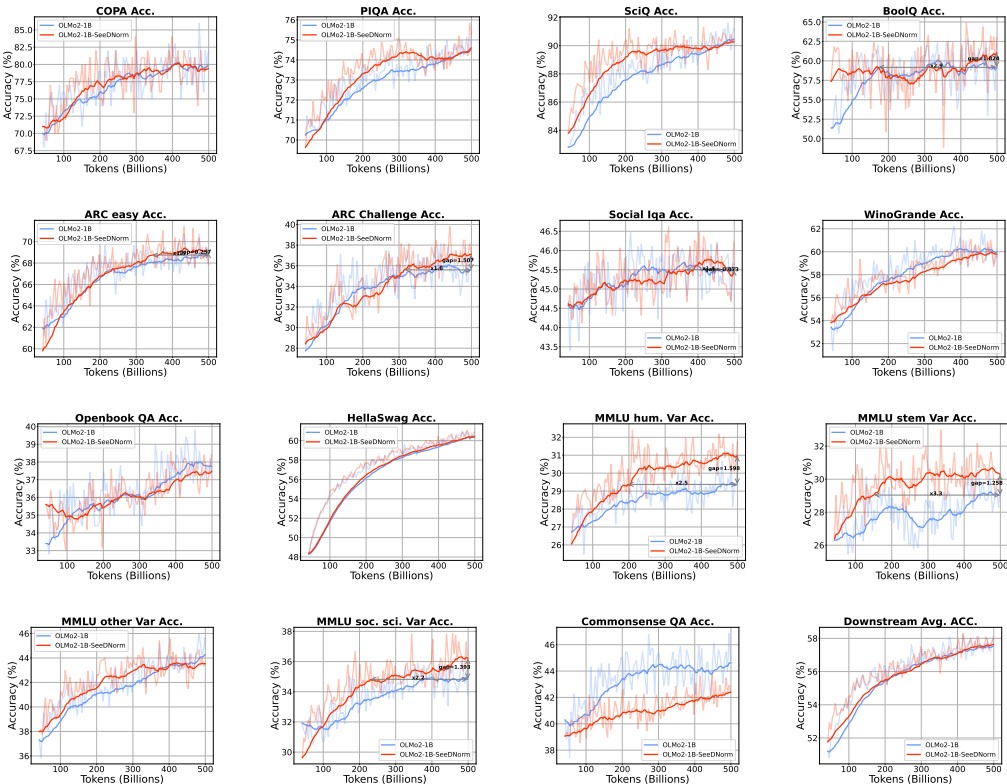

Figure 10: Comparisons of the accuracy of all downstream tasks from Table 5 in **OLMo2-1B** when using SeeDNorm as the normalization layer versus the default RMSNorm. The figure illustrates the evolution of downstream task accuracy as the total training tokens increase during training, with transparent lines indicating unsmoothed results and solid lines denoting 0.99 EMA-smoothed results.

capabilities, necessitating a further increase in its drop path rate to 0.6. For all models, the final classification accuracy is evaluated using the EMA model.

Table 10: Configuration and hyperparameters for **ViT-B** and **ViT-L**.

| Model | ViT-B | ViT-L |
|---|---|---|
| Num Layer | 12 | 24 |
| Hidden Dim | 768 | 1024 |
| Num Head | 12 | 16 |
| Patch Size | $16 \times 16$ | |
| EMA | 0.9999 | 0.9999 |
| Input Resolution | $224 \times 224$ | |
| DropPath | 0.1 | 0.6 |
| Global Pool | Average Pooling | |
| SeeDNorm Dropout | Yes | Yes |
| SeeDNorm Head | 16 | 32 |

Table 11: Hyperparameters for the optimizer of **ViT-B** and **ViT-L** in image classification task.

| Model | ViT-B | ViT-L |
|---|---|---|
| Optimizer | AdamW($\beta_1 = 0.9, \beta_2 = 0.999$) | |
| Learning Rate | 4e-3 | 4e-3 |
| Weight Decay | 0.05 | 0.1 |
| LR Schedule | Cosine Schedule | |
| Warm up | 20 Epochs | |
| Gradient Clip | 1.0 | 1.0 |
| Global Batch Size | 4096 | 4096 |
| Training Epochs | 300 | 300 |

Table 12: Configuration and hyperparameters for **ConvNeXT-B** and **ConvNeXT-L**.

| Model | ConvNeXT-B | ConvNeXT-L |
|---|---|---|
| Depth | [3, 3, 27, 3] | [3, 3, 27, 3] |
| Dims | $\begin{bmatrix} 128 \\ 256 \\ 512 \\ 1024 \end{bmatrix}$ | $\begin{bmatrix} 192 \\ 384 \\ 768 \\ 1536 \end{bmatrix}$ |
| EMA | 0.9999 | 0.9999 |
| Input Resolution | $224 \times 224$ | |
| DropPath | 0.5 | 0.5 |
| SeeDNorm Dropout | No | No |
| SeeDNorm Head | 16 | 32 |
| ls init value | 1e-6 | 1e-6 |
| head init scale | 1.0 | 1.0 |

Table 13: Hyperparameters for the optimizer of **ConvNeXT-B** and **ConvNeXT-L** in image classification task.

| Model | ConvNeXT-B | ConvNeXT-L |
|---|---|---|
| Optimizer | AdamW($\beta_1 = 0.9, \beta_2 = 0.999$) | |
| Learning Rate | 4e-3 | 4e-3 |
| Weight Decay | 0.05 | 0.1 |
| LR Schedule | Cosine Schdule | |
| Warm up | 20 Epochs | |
| Gradient Clip | 1.0 | 1.0 |
| Global Batch Size | 4096 | 4096 |
| Training Epochs | 300 | 300 |

The model configurations and hyper-parameters in our used ConvNeXT-B and ConvNeXT-L models in image classification task are presented in Table 12. The training and optimization parameters for both models are shown in Table 13. The configuration of ConvNeXT is largely consistent with that of ViT. The main difference lies in the ConvNeXT model, which is structured into four stages, each with varying depths and dimensions. The detailed configuration is presented in Table 12.

### D.2.1 EXPERIMENT RESULTS

In the main text, we have already reported the accuracy results for the image classification task. Figure 11 and Figure 12 present detailed comparisons of the loss curves during training. With the application of SeeDNorm, our method demonstrates a clear advantage over DyT in the training curves. When scaling up to ViT-L, despite using a higher drop path rate, the loss advantage becomes even more pronounced. This also reflects an enhanced fitting capability of the model under the influence of SeeDNorm, although this gap is not effectively reflected in the accuracy data on the test set.

### D.3 ViT IN MAE

In the MAE pre-training and fine-tuning task, we also employed ViT-B and ViT-L as the primary research models. The model structures of ViT-B and ViT-L are consistent with those presented in Table 10, with the distinction that neither EMA is used during the pre-training nor the subsequent fine-tuning processes for MAE, drop path and dropout is also not used in the pre-training phase. The training and optimizer configurations for MAE pre-training and fine-tuning are detailed in Table 14 and Table 15, respectively. During the fine-tuning process, for ViT-L, we increase the drop path rate to further prevent overfitting. For the remaining configurations, we maintained consistency with the DyT (Zhu et al., 2025b) baseline.

### D.3.1 EXPERIMENT RESULTS

In Figure 13, we plot the loss comparison curves during the pre-training process. Whether for ViT-B or ViT-L, the application of SeeDNorm significantly reduces the loss. In Figure 14, we illustrate the

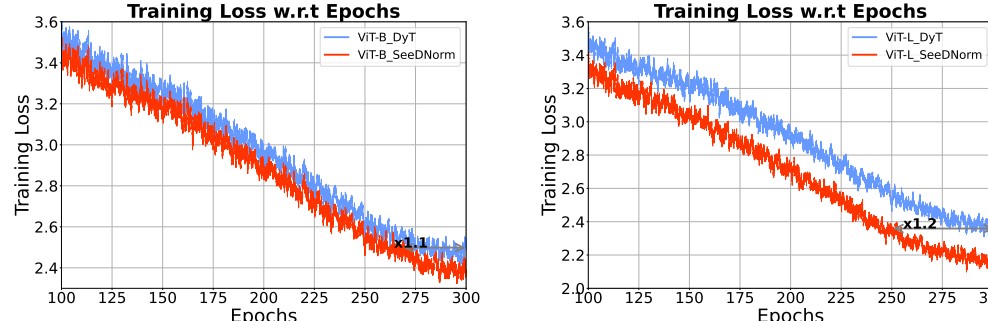

Figure 11: Comparison of the Cross Entropy loss curves for ViT-B and ViT-L on the ImageNet image classification task using DyT and SeeDNorm, plotted against the number of training epochs. The loss curves have been smoothed using a 0.99 EMA.

loss variation curves during the fine-tuning process, where SeeDNorm similarly achieves a notable reduction in loss during fine-tuning.

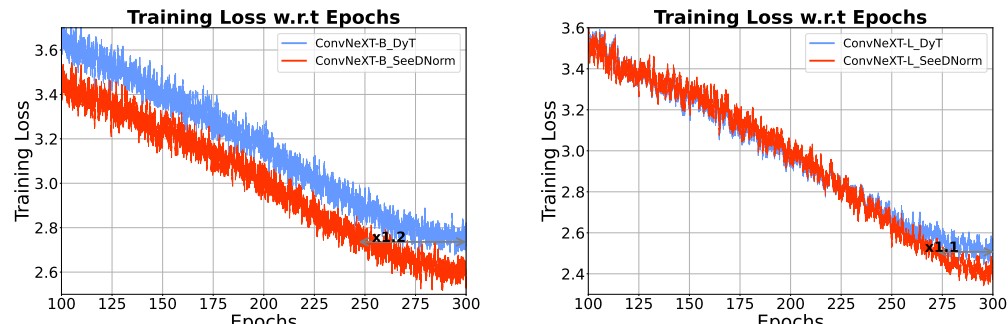

Figure 12: Comparison of the Cross Entropy loss curves for ConvNeXT-B and ConvNeXT-L on the ImageNet image classification task using DyT and SeeDNorm, plotted against the number of training epochs. The loss curves have been smoothed using a 0.99 EMA.

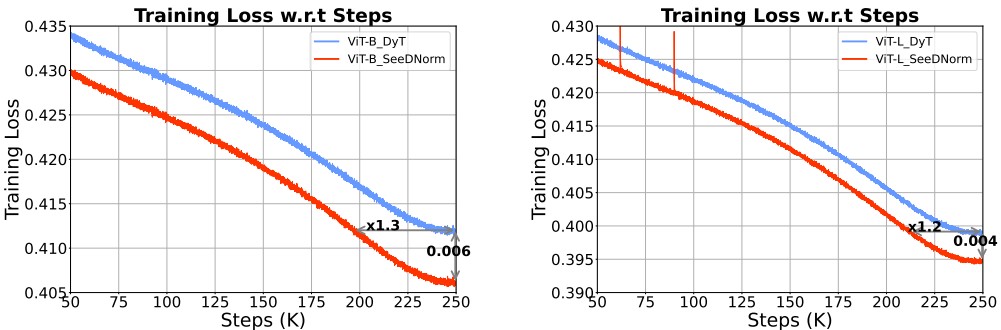

Figure 13: Comparison of the MSE loss curves for ViT-B and ViT-L on the MAE self-supervised image masking reconstruction task using DyT and SeeDNorm, plotted against the number of training epochs. The loss curves have been smoothed using a 0.99 EMA.

## D.4 DIT IN IMAGE GENERATION

In the image generation task, we conducted experiments based on DiT-B (Peebles & Xie, 2023) and DiT-XL (Peebles & Xie, 2023), with the model configurations detailed in Table 16 and training hyperparameters in Table 17, respectively. For the image generation task, because the random noise

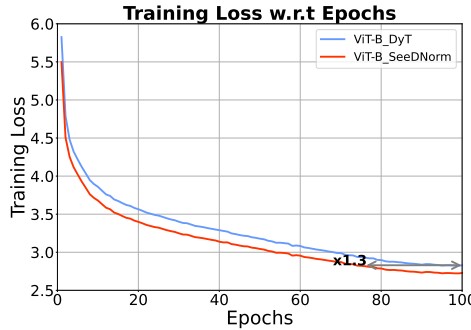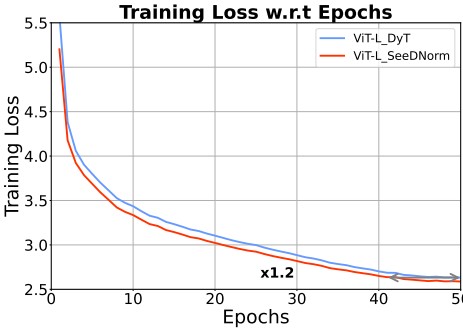

Figure 14: Comparison of the Cross Entropy loss curves against the number of training epochs for ViT-B and ViT-L, using DyT and SeeDNorm with full-parameter fine-tuning initialized with MAE pre-trained weights on the ImageNet image classification task.

and timestep sampling of diffusion greatly enrich sample diversity, it is harder for the model to overfit compared to image classification tasks, and using the standard form of SeeDNorm is sufficient to ensure stable training of the model. Therefore, we have not explored the multi-head form at this stage.

Table 14: Hyperparameters for the optimizer of **ViT-B** and **ViT-L** in MAE pre-training.

| Model | ViT-B | ViT-L |
|---|---|---|
| Optimizer | AdamW ($\beta_1 = 0.9, \beta_2 = 0.95$) | |
| Learning Rate | 2.4e-3 | 2.4e-3 |
| LR Schedule | Cosine Schedule | |
| Weight Decay | 0.05 | 0.05 |
| Mask Ratio | 0.75 | 0.75 |
| Warm up | 40 Epochs | |
| Gradient Clip | No | No |
| Global Batch Size | 4096 | 4096 |
| Training Epochs | 800 | 800 |

Table 15: Hyperparameters for the optimizer of **ViT-B** and **ViT-L** in MAE fine-tuning.

| Model | ViT-B | ViT-L |
|---|---|---|
| Optimizer | AdamW ($\beta_1 = 0.9, \beta_2 = 0.999$) | |
| Learning Rate | 2e-3 | 4e-3 |
| LR Schedule | Cosine Schedule | |
| Weight Decay | 0.05 | 0.05 |
| DropPath | 0.1 | 0.2 |
| Warm up | 5 Epochs | 5 Epochs |
| Gradient Clip | No | No |
| Global Batch Size | 1024 | 1024 |
| Training Epochs | 100 | 50 |

Table 16: Configuration and hyperparameters for **DiT-B/4** and **DiT-XL/2**.

| Model | DiT-B/4 | DiT-XL/2 |
|---|---|---|
| Num Layer | 12 | 28 |
| Hidden Dim | 768 | 1152 |
| Num Head | 12 | 16 |
| Image Size | $256 \times 256$ | $256 \times 256$ |
| Latent Size | $32 \times 32$ | $32 \times 32$ |
| Patch Size | 4 | 2 |
| MLP Ratio | 4 | 4 |

Table 17: Hyperparameters for the optimizer of **DiT-B/4** and **DiT-XL/2** in image classification task.

| Model | DiT-B/4 | DiT-XL/2 |
|---|---|---|
| Optimizer | AdamW ($\beta_1 = 0.9, \beta_2 = 0.999$) | |
| Learning Rate | 1e-4 | 1e-4 |
| LR Schedule | Constant | |
| Weight Decay | - | - |
| class drop prob | 0.1 | 0.1 |
| Global Batch Size | 256 | 256 |

# E PYTORCH IMPLEMENTATION OF SEEDNORM

In Algorithm 1 and Algorithm 2, we implement our proposed SeeDNorm and its Multihead form respectively, using a PyTorch-like style.

---

**Algorithm 1** Pseudocode of SeeDNorm in a PyTorch-like style.

---

```
class SeeDNorm(Module):
  def __init__(self, D, init):
    super().__init__()
    self.α = Parameter(ones(D) * init)
    self.β = Parameter(zeros(D))
    self.γ = Parameter(ones(D))

  def forward(self, x):
    rescale = tanh(x @ self.β)
    x = x / RMS(x)
    dynamic_scale = rescale.unsqueeze(1) * self.α
    return (dynamic_scale + self.γ) * x
```

---

**Algorithm 2** Pseudocode of Multihead SeeDNorm in a PyTorch-like style.

---

```
class SeeDNorm(Module):
  def __init__(self, D, init, num_heads):
    super().__init__()
    self.α = Parameter(ones(D) * init)
    self.β = Parameter(zeros(D))
    self.γ = Parameter(ones(D))
    self.num_heads = num_heads

  def forward(self, x):
    B, N, D = x.shape
    x_dtype = x.dtype
    h = x.reshape(B, N, self.num_heads, D // self.num_heads).transpose(1, 2)
    β = self.β.reshape(1, self.num_heads, 1, D // self.num_heads).repeat(B, 1, 1, 1).transpose(−1, −2)
    activate = tanh(torch.matmul(h, β).float())
    α = self.α.reshape(1, self.num_heads, 1, D // self.num_heads).repeat(B, 1, 1, 1)
    dynamic_scale = activate * α
    dynamic_scale = dynamic_scale.to(x_dtype)
    x = x / RMS(x)
    return (dynamic_scale + self.γ) * x
```

---

# F  NUMBER OF PARAMETERS, COMPUTATIONAL COST AND LATENCY

## F.1  THEORETICAL ANALYSIS

Compared to RMSNorm, SeeDNorm introduces two additional $D$-dimensional parameters, $\boldsymbol{\alpha}$ and $\boldsymbol{\beta}$. In the entire Transformer network, assuming each layer includes input normalization for the attention layer and FFN, QKNorm within the attention layer, and normalization at the Transformer output, the newly introduced parameters amount to $(2 \times 2D + 2 \times 2 \times D/H) \times N + 2D = (4N + 4\frac{N}{H} + 2) \times D$, where $H$ is the number of attention heads. Since $N$ is much smaller than $D$, the increase in the overall parameter is much smaller than a linear layer, which can be considered negligible. In terms of computational complexity, compared to RMSNorm, SeeDNorm introduces two additional matrix multiplications, one element-wise activation, and one element-wise addition along the channel dimension. For each token $\boldsymbol{x} \in \mathbb{R}^{1 \times D}$, compared to RMSNorm, the number of additional multiplications is $2D$, and the number of additional additions is $D + D - 1$. The total additional multiply-add operations for the entire network are $(4D-1) \times (2N+1) + (\frac{4D}{H} - 1) \times 2N = (8N + \frac{8N}{H} + 4) \times D - 4N - 1 = O(D)$. In contrast, a $D \times D$ linear layer already involves a computational complexity of $O(D^2)$. Therefore, the additional computational overhead introduced by SeeDNorm remains negligible.

---

**Algorithm 3** Triton Implementation of the forward process of SeeDNorm.

---

```
@triton.jit
def seednorm_fwd_kernel(
    X, Y, W, alpha, beta, stride_ml, stride_n, L, N, eps, BLOCK_SIZE: tl.constexpr,
):
    row = tl.program_id(0)
    batch = tl.program_id(1)

    base_idx = row * stride_ml + batch * stride_n
    Y += base_idx
    X += base_idx

    _rms = tl.zeros([BLOCK_SIZE], dtype=tl.float32)
    _dot_product = tl.zeros([BLOCK_SIZE], dtype=tl.float32)
    for off in range(0, N, BLOCK_SIZE):
        cols = off + tl.arange(0, BLOCK_SIZE)
        a = tl.load(X + cols, mask=cols < N, other=0.0).to(tl.float32)
        beta_element = tl.load(beta + cols, mask=cols < N).to(tl.float32)
        _rms += a * a
        _dot_product += a * beta_element

    rms = tl.sqrt(tl.sum(_rms) / N + eps)
    dot_product = tl.sum(_dot_product)
    neg_two_x = -2.0 * dot_product
    exp_neg_two_x = tl.exp(neg_two_x)
    dot_product = (1.0 - exp_neg_two_x) / (1.0 + exp_neg_two_x)

    for off in range(0, N, BLOCK_SIZE):
        cols = off + tl.arange(0, BLOCK_SIZE)
        mask = cols < N
        w = tl.load(W + cols, mask=mask)
        alpha_element = tl.load(alpha + cols, mask=mask)
        x = tl.load(X + cols, mask=mask, other=0.0).to(tl.float32)
        x_hat = x / rms
        y = x_hat * (w + alpha_element * dot_product)
        tl.store(Y + cols, y.to(X.dtype.element_ty), mask=mask)

class SeeDNorm(Module):
    def __init__(self, D, init):
        super().__init__()
        self.α = Parameter(ones(D) * init)
        self.β = Parameter(zeros(D))
        self.γ = Parameter(ones(D))

    def forward(x):
        y = torch.empty_like(x)
        M, L, N = x.shape
        grid = (M, L)
        seednorm_fwd_kernel[grid](
            x, y, self.weight, self.alpha, self.beta, x.stride(0), x.stride(1), L, N, self.eps, BLOCK_SIZE=1024
        )
        return y
```

---

However, when using only the PyTorch implementation, SeeDNorm requires more memory access operations and these operations are more fragmented compared to RMSNorm. This will affect latency and overall efficiency to a certain extent. In practical applications, we recommend fusing the operations into a single kernel function, thereby achieving comparable efficiency. The implementation of Triton kernel of the forward process is shown in Algorithm 3.

Table 18: Latency comparison between SeeDNorm and RMSNorm, which are implemented in PyTorch and Triton respectively, when processing a single-token batch and a single batch with 1024 tokens. The dimension of all tokens is uniformly set to 1536.

| Module | | SeeDNorm | RMSNorm |
|---|---|---|---|
| **PyTorch** | 1 Token | $1.870 \times 10^{-4}$ | $6.049 \times 10^{-5}$ |
| **Latency (s)** | 1024 Tokens | $1.920 \times 10^{-4}$ | $6.427 \times 10^{-5}$ |
| **Triton** | 1 Token | $2.859 \times 10^{-5}$ | $2.829 \times 10^{-5}$ |
| **Latency (s)** | 1024 Tokens | $2.836 \times 10^{-5}$ | $2.833 \times 10^{-5}$ |

### F.2 MODULE-LEVEL STANDALONE LATENCY

The overall training speed of a model can be influenced by many external factors, such as I/O and CPU usage, which limits the interpretability of this metric. Moreover, since normalization layers inherently account for only a very small portion of the total computation, their impact on end-to-end training speed is minimal. Therefore, we conduct a runtime efficiency analysis focused solely on the normalization module itself. In Table 18, we report the forward-pass latency of the SeeDNorm module compared to the RMSNorm module. The reported values represent the average latency over 1000 forward passes, excluding the first 10 iterations, since PyTorch may compile certain operators during initial execution.

As shown in Table 18, if implemented purely with PyTorch, SeeDNorm exhibits higher latency than RMSNorm. As explained in Section F.1, this is because a vanilla PyTorch implementation involves relatively more frequent global memory reads and writes, leading to non-negligible memory access overhead. When optimized with Triton, SeeDNorm enables the fusion of the dynamic term computation into the computational pipeline of RMSNorm. As a result, it achieves nearly identical latency compared with the standard RMSNorm module, while consistently preserving its computational efficiency across both single-token batches and 1024-token batches.

