# OpenReview forum: "SeeDNorm: Self-Rescaled Dynamic Normalization"
_ICLR.cc/2026/Conference — ICLR 2026 Poster_

### Official Review · Reviewer_NMms · 2025-10-25

**Soundness:** 3
**Presentation:** 3
**Contribution:** 3
**Rating:** 6
**Confidence:** 3

**Summary:**

This paper proposes a new normalization method called SeeDNorm.
The purpose of this normalization method is to address the wide-ranging variations and distribution shifts in input data that a static scaling factor γ cannot handle.
The proposed method is simple yet effective.
Its effectiveness is discussed in the context of large-scale language model pre-training, as well as supervised and unsupervised vision tasks (such as image generation).

**Strengths:**

- The paper is well-structured and easy to understand.
  - It strikes a good balance between theoretical explanation and experimental validation.
- Experiments are conducted on both language models and vision tasks, discussing effectiveness across a broad range of domains.
- References are appropriately cited.

**Weaknesses:**

- I consider this a well-written paper. None of these points represent fatal flaws as a paper; they are comments to better understand its potential.

- While evaluating results on OLMo2 as a language model, demonstrating effectiveness on other language models would further strengthen the paper's persuasiveness.

- Regarding vision models, effectiveness has been confirmed for CNN-based models like ConvNext. I consider this a very positive point. However, for CNN-based models, techniques beyond Batch Norm exist, such as GroupNorm, Weight Normalization, and Weight Standardization. Is comparison or combined use with these possible?

**Questions:**

- I fundamentally believe this is a well-written paper. These are questions to better understand the paper's potential. Please see the weaknesses section for details.
  - Is the proposed method effective with language models other than OLMo2?
  - Regarding the possibility of combining it with other normalization techniques, if you already have insights, please share them.
  - Is the implementation planned to be released?

---

> ### Author Response · Authors · 2025-11-25
> **Response to Reviewer NMms**
>
> Thank you very much for your valuable comments and positive feedback on our paper. Below are our responses:
>
> ---
>
> > W1: While evaluating results on OLMo2 as a language model, demonstrating effectiveness on other language models would further strengthen the paper's persuasiveness..
> >
>
> `Response:`In addition to conducting experiments on OLMo2 of different sizes 550M and 1B, we also performed experiments on the mixture-of-experts (MoE)-based large language model OLMoE with both 1B and 7B parameter sizes. Our method achieves even more pronounced performance gains in the MoE architecture, as referenced in Table 1 of our main text and our response to Reviewer `EBT8` `W1`.
>
> ---
>
> > W2: Regarding vision models, effectiveness has been confirmed for CNN-based models like ConvNext. I consider this a very positive point. However, for CNN-based models, techniques beyond Batch Norm exist, such as GroupNorm, Weight Normalization, and Weight Standardization. Is comparison or combined use with these possible?
> >
>
> `Response:` Thank you for your feedback. We fully understand your expectation that SeeDNorm be extended to more types of normalization layers. Our proposed SeeDNorm is primarily designed to be applied to **the currently dominant Transformer architectures** as well as network structures that use token-wise LayerNorm or RMSNorm, such as ConvNeXt. By contrast, Batch Normalization (BN) and Group Normalization (GN) operate along the batch dimension (BN for full batch, GN for intra-sample hidden dimension groups) — their fundamental computation paradigms (including the scope of statistic collection and the dimension of normalization operations) differ entirely from SeeDNorm’s. This discrepancy means our current theoretical analysis of "token-wise dynamic scaling stability" cannot be directly applied to batch/group-based normalization layers.
>
> However, to explore the generality of our dynamic scaling idea, now we conduct experiments to introduce SeeDNorm-inspired dynamic adjustment mechanisms into GN (hereafter tentatively named "SeeD-GN"). Similarly, SeeD-GN adjusts the scaling coefficient $\pmb{\gamma}$ in GN as $\sigma(\pmb{x} · \pmb{\beta}^T) · \pmb{\alpha} + \pmb{\gamma}$. We adopted exactly the same experimental parameters as ViT-B in our paper, including the number of heads in SeeD-GN. But we do not use EMA for model updates and directly utilized the model itself. Performance results are presented in the table below— the dynamic adjustment method of SeeDNorm still yielded performance improvements in the GN setting.
>
> | **Model** | **ResNet50-GN** | **ResNet50-SeeD-GN** |
> | --- | --- | --- |
> | Acc@1 | 78.5 | 78.6 |
> | Acc@5 | 94.2 | 94.4 |
>
> However, this experiment serves only as a preliminary reference, and we have not conducted a rigorous theoretical analysis on Group Normalization. The gradient and training stability are not analysed as well. We are also highly interested in conducting further research on this topic in the future.
>
> ---
>
> > Q1: Is the proposed method effective with language models other than OLMo2?
> >
>
> `Response:` We have responsed this question in our reply to `W1` .
>
> ---
>
> > Q2: Regarding the possibility of combining it with other normalization techniques, if you already have insights, please share them.
> >
>
> `Response:`We have responsed this question in our reply to `W2` .
>
> ---
>
> > Q3: Is the implementation planned to be released?.
> >
>
> `Response:` Yes, we plan to release more detailed source code. Additionally, we have included the Triton forward kernel implementation for SeeDNorm in both the revised manuscript and our unified response “**Unified Response to the Computational Overhead and Runtime Efficiency of SeeDNorm**” to the reviewers.

---

> > ### Comment · Reviewer_NMms · 2025-11-26
> > **Comments for Authors**
> >
> > Dear Authors,
> >
> >
> > First and foremost, thank you for your thoughtful response.
> > As I indicated in my initial review, I have a positive assessment of this paper.
> > I also appreciate your discussion regarding the combination with GN.
> > Thank you for sharing your perspective on the code policy as well.
> > I believe your insightful and expert comments will serve as invaluable reference material for the AC when making their decision.
> > Finally, it has been an honor to learn so much from researchers and reviewers with such high levels of expertise.
> >
> >
> > Thank you very much.

---

### Official Review · Reviewer_gTBE · 2025-10-31

**Soundness:** 3
**Presentation:** 3
**Contribution:** 3
**Rating:** 6
**Confidence:** 4

**Summary:**

This paper introduces SeeDNorm, a novel normalization layer for neural networks. SeedNorm is an extension of RMSNorm, that modulates the scale parameter gamma of RMSNorm by adding a term to gamma which consists of two learnable parameters. These parameters act on the unnormalized input of the normalization layer, and thus allow scale information to flow through the normalization layer. The authors prodide a gradient analysis of SeeDNorm in order to show that their normalization layer is insensitive to extremely large or small values, and to justify their regularization and initialization choices. They provide experiments with OLMo models for LLM pretraining, DiTs for Image Generation, ViTs and ConvNext for Image Classification, and ViTs for Self-supervised learning, overall reporting improvements over baselines like LayerNorm and DyT. They also provide ablations on their design choices.

**Strengths:**

- I appreciate the large scope of the experiments: The paper presents extensive studies across tasks and scale, spanning LLM pretraining (with normal and MoE models), image classification, and image generation.
- SeeDNorm either matches or outperforms other normalization variants. The strongest benefits are observed for MoE language models
- The ablations are interesting and comprehensive. They answer several questions that came up when reading through the paper
- the paper is well-written and easy to follow

**Weaknesses:**

- A few experimental details are missing, and it is unclear if default training scripts or modified scripts have been used (more below in questions)
- For Vision experiments, especially image classification, the benefits of SeeDNorm are less clear. In Table 2, the numbers are either identical or extremely close to baselines (yet the SeedNorm variant is bold - why?). Further, the default SeedNorm variant fails to converge (Table 3), and Multihead SeedNorm is required to match or bring gains over LayerNorm or DyT, with different choices of head dimension often leading to suboptimal performance. Also the drop path rate had to be adjusted for large models “to further prevent overfitting”. This questions if SeeDNorm can be used as an easy drop-in replacement across tasks
- as for the vision experiments, the numbers for OLMo are sometimes close. Since there are no error bars or runs over multiple seeds (which is partially understandable, given the computational cost of large scale experiments), it is hard to judge the significance of the results
- In Theorem 3.2 the authors claim that the dot product of two random vectors is inversely proportional to their dimension D. However, they then proceed to show that it is proportional (not inversely) to D. If it were indeed inversely proportional, splitting the vectors to reduce the variance would not make sense

Minor remarks:
- typo in line 28, neglligible
- line 1110: “Our method similarly achieved consistent improvements” ( OLMo2-1B), but this varies across tasks, and on average, the accuracy is almost identical
- in the abstract in line 18, the connection between a static scaling factor zero-shot setups (as compared to other setups) is not clear to me

**Questions:**

- How was the training setup for the LLM experiments chosen? Is it the default OLMo recipe?
- in line 302 the authors write that the parameter $\alpha$ is initialized to 1 for language experiments. I could not find if it is initialized differently for vision experiments. Can the authors clarify this?
- How was the weight decay for $\alpha,\beta$ chosen? Was it the default weight decay used for other parameters, or was it tuned?
- the numbers in Table 2 are exactly identical to the numbers in the DyT paper. Is this because the authors used the same codebase? Or were the numbers taken from that paper?
- Overall, was there any form of hyperparameter tuning or adaption involved for SeedNorm, or were always the default recipes taken from somewhere (where?), and only the normalization layer was replaced?

---

> ### Author Response · Authors · 2025-11-25
> **Response to Reviewer gTBE (Part1)**
>
> We sincerely appreciate your detailed evaluation and constructive comments on our manuscript. Below are our responses, which we hope will satisfactorily address your concerns.
>
> ---
> > W1: A few experimental details are missing, and it is unclear if default training scripts or modified scripts have been used.
> >
> `Response:`We sincerely appreciate your question. From the perspective of method comparison, we strictly controlled all model configurations except for the normalization layer to remain identical—thus this might not be a major weakness of our experimental design. Below, we clarify the specific configurations:
> - For OLMoE‑7B, we used the official default configuration. For OLMoE‑1B, the parameter count is 1.3B. We adjusted the number of layers and the hidden dimension so that the total parameters match 1.3B, as this size is a well‑recognized “sweet spot.” It is widely considered the smallest scale at which one can observe initial emergent abilities; models such as LLaMA and GPT have also been trained at this scale. At around 1.3B parameters, models typically begin to exhibit in‑context learning, basic reasoning, and instruction‑following capabilities. Moreover, 1.3B is also close to the largest model size that can be both trained and deployed on a single consumer‑grade or entry‑level professional GPU (e.g., RTX 4090 or V100). For these reasons, we trained a model at the 1.3B scale as well.
> - For OLMo2, the OLMo2‑1B configuration follows the official defaults, whereas OLMo2‑550M is a smaller model we configured specifically to enable faster iteration.
>
> ---
> > W2: For Vision experiments, especially image classification, the benefits of SeeDNorm are less clear. In Table 2, the numbers are either identical or extremely close to baselines (yet the SeedNorm variant is bold - why?).
> >
>
> `Response:` We sincerely appreciate your comment, and we would like to respectfully clarify that many of the observed improvements are in fact not subtle. The experimental results presented in Table 2 are obtained from training and testing on ImageNet-1K. It is widely acknowledged that **further performance improvements on ImageNet-1K are inherently limited** due to long-standing issues with the dataset itself, such as annotation errors, class imbalance, and variations in sample quality. For example, compared to Swin Transformer Small [1], Swin Transformer Base  increases the parameter count by 66% and the computational overhead by nearly 80%, yet it only achieves a marginal 0.5% improvement in Top-1 accuracy on ImageNet-1K. In contrast, our proposed SeeDNorm introduces **negligible parameter increment and computational overhead (as detailed in our unified response “Unified Response to the Computational Overhead and Runtime Efficiency of SeeDNorm” to the reviewers),** With this near-zero additional cost, we directly achieved a Top-1 accuracy improvement of approximately 0.5% for both ViT-B and ViT-L models on ImageNet-1K. Considering the strict resource constraints of this "near-free lunch" setting, we believe this improvement could not be deemed subtle.
>
> The bold text formatting used in the tables is primarily intended to highlight and distinguish the experimental results obtained using our proposed SeeDNorm method from those of other baseline or comparative methods.
>
> [1] Liu Z, Lin Y, Cao Y, et al. Swin transformer: Hierarchical vision transformer using shifted windows[C]//Proceedings of the IEEE/CVF international conference on computer vision. 2021: 10012-10022.

---

> ### Author Response · Authors · 2025-11-25
> **Response to Reviewer gTBE (Part2)**
>
> ---
>
> > W2: If SeeDNorm can be used as an easy drop-in replacement across tasks
> >
>
> `Response:` SeeDNorm can be integrated into diverse tasks and model architectures in a highly straightforward and simplified manner. In fact, except for ImageNet classification—where models are trained for many epochs on a relatively small dataset and thus prone to overfitting—no special modifications are needed. For this specific setting, we adopt a multi‑head version of SeeDNorm and adjust the drop‑path rate to mitigate overfitting. Beyond this case, SeeDNorm can be applied directly across tasks without altering other hyperparameters. We keep all other settings unchanged, such as using $\pmb{\alpha} = \pmb{1}$ as initial value and applying the same weight decay as the rest of the network, and we observe consistent improvements across all tested tasks.
>
> In other words, the additional adjustments we made are solely aimed at combating overfitting and reducing gradient variance. The fact that overfitting becomes easier to trigger also indirectly indicates that SeeDNorm enhances the model’s representational capacity.
>
> In addition, integrating SeeDNorm into existing network architectures is extremely straightforward: one simply replaces all token‑wise normalization layers (such as LayerNorm or RMSNorm) in the model with SeeDNorm, without requiring any further architectural changes. As discussed in our response to Reviewer **`EBT8` `W3`**, we also evaluated SeeDNorm on detection and segmentation tasks, where it likewise leads to consistent performance improvements.
>
> ---
>
> > W3: As for the vision experiments, the numbers for OLMo are sometimes close. Since there are no error bars or runs over multiple seeds (which is partially understandable, given the computational cost of large scale experiments), it is hard to judge the significance of the results
> >
>
> `Response:` We sincerely appreciate your suggestion. First, we wish to clarify a potential misunderstanding: OLMo is a model family designed for **language modeling tasks**, not vision experiments.
>
> Regarding the issue of experimental reproducibility with large-scale language models: While we acknowledge that running multiple rounds of experiments with different random seeds for large LLMs is extremely resource-intensive (due to their massive parameter counts and lengthy training durations), this does not make our results “hard to judge”. All of our LLM experiments were conducted with **fixed, identical random seeds**—the exact same ones provided in the official open-source codebases of OLMoE and OLMo2. This strict seed consistency eliminates the interference of random seed variations on experimental outcomes and ensures that the performance gains we observed stem entirely from the effectiveness of our proposed SeeDNorm method, rather than fluctuations caused by random initialization or training processes. As such, our results are reliable and reproducible.
>
> ---
>
> > W4: In Theorem 3.2 the authors claim that the dot product of two random vectors is inversely proportional to their dimension D. However, they then proceed to show that it is proportional (not inversely) to D. If it were indeed inversely proportional, splitting the vectors to reduce the variance would not make sense
> >
>
> `Response:`We sincerely appreciate you pointing out our typo. The title of Theorem 3.2 should be stated "the dot product of two random vectors is proportional to their dimension ", and we have corrected this error in the revised version of the manuscript.
>
> ---
>
> > W5: typo in line 28, neglligible
> >
>
> `Response:`Thank you for your careful reading and for pointing out the typo. We have now corrected it.
>
> ---
>
> > W6: “Our method similarly achieved consistent improvements” ( OLMo2-1B), but this varies across tasks, and on average, the accuracy is almost identical
> >
>
> `Response:`Thank you for the correction. OLMo2‑1B should indeed be described as achieving improvements on *most* downstream tasks, rather than all. We have updated our wording accordingly. The average accuracy is primarily lowered by the performance on CommonsenseQA, which reduces the overall gap.

---

> ### Author Response · Authors · 2025-11-25
> **Response to Reviewer gTBE (Part3)**
>
> ---
>
> > W7: In the abstract in line 18, the connection between a static scaling factor zero-shot setups (as compared to other setups) is not clear to me
> >
>
> `Response:`We would be very happy to clarify this point. The term “static scaling factor” refers to the weight used in RMSNorm, which scales each channel with a fixed parameter that does not depend on the input. In other words, all input vectors are scaled in exactly **the same way**. This static weight is forced to **learn an average optimal scaling over all training tasks**. However, the input distribution encountered in zero‑shot settings can differ from the training data, and such a static weight cannot adapt to the statistical characteristics of these unseen distributions (e.g. downstream tasks in Table 5).
>
> ---
>
> > Q1: How was the training setup for the LLM experiments chosen? Is it the default OLMo recipe?
> >
>
> `Response:`We have already answered this question in our response section specifically targeting `W1`.
>
> ---
>
> > Q2: In line 302 the authors write that the parameter is initialized to 1 for language experiments. I could not find if it is initialized differently for vision experiments. Can the authors clarify this?
> >
>
> `Response:`We sincerely appreciate your valuable feedback. In all our experiments **except for the ablation studies explicitly examining the impact of** $\alpha$, we uniformly set $\alpha$  as the default value.
>
> While we acknowledge that there may exist an optimal $\alpha$ configuration tailored to each specific task, we did not perform meticulous hyperparameter tuning for $\alpha$ across all tasks. This intentional design choice and the consistent performance gains we observed also highlight the **robustness and general applicability** of our proposed SeeDNorm.
>
> ---
>
> > Q3: How was the weight decay for chosen? Was it the default weight decay used for other parameters, or was it tuned?
> >
>
> `Response:` The weight decay applied to SeeDNorm is set to the same value as that used for the overall network. Similar to the parameter $\alpha$, we did not perform any additional task-specific or component-specific tuning for this hyperparameter.
>
> ---
>
> > Q4: The numbers in Table 2 are exactly identical to the numbers in the DyT paper. Is this because the authors used the same codebase? Or were the numbers taken from that paper?
> >
>
> `Response:` All data for DyT presented in the Table 2 are directly cited from the original DyT paper. Additionally, we also used the exact same codebase as provided in the original DyT implementation for verification, and the accuracy we reproduced in our experimental environment is either consistent with or slightly lower than the reported values (with an absolute error within 0.1% Acc@1).
>
> ---
>
> > Q5: Overall, was there any form of hyperparameter tuning or adaption involved for SeedNorm, or were always the default recipes taken from somewhere (where?), and only the normalization layer was replaced?
> >
>
> `Response:` We have already answered this question in our response to *"W2: If SeeDNorm can be used as an easy drop-in replacement across tasks"*.
>
> To recap briefly:
>
> - **General applicable scenarios**: SeeDNorm can be directly applied as a drop-in replacement for all token-wise normalization layers (LayerNorm/RMSNorm) in existing networks without any other structural adjustments.
> - **Scenario-specific targeted adjustments (only for extreme overfitting cases)**: For tasks prone to overfitting—such as ImageNet-1K classification (trained on a relatively small dataset for many epochs)—minimal targeted adjustments can be made to further stabilize training:
>     1. Use the multi-head variant of SeeDNorm, with the head dimension typically set less than 100 (such as 32, 64 and 96).
>     2. Increase the drop path rate.
>
>     All other hyperparameter configurations can remain entirely unchanged in both scenarios.

---

### Official Review · Reviewer_HBqF · 2025-11-01

**Soundness:** 4
**Presentation:** 3
**Contribution:** 3
**Rating:** 8
**Confidence:** 4

**Summary:**

This paper proposes SeeDNorm (Self-Rescaled Dynamic Normalization), a dynamic normalization method that preserves input norm information and adaptively adjusts the scaling factor based on the current input. The work aims to address the limitation of RMSNorm, which discards the input norm during the forward pass and relies on a fixed scaling factor that cannot adapt to data variability or distributional shifts, leading to suboptimal performance in zero-shot scenarios. SeeDNorm introduces a self-rescaling mechanism that modulates the scaling coefficient according to the input itself, thereby retaining norm information and enabling data-dependent normalization. The paper provides theoretical analysis of scaling invariance and gradient behavior in both forward and backward propagation, showing that SeeDNorm maintains numerical stability and adaptively adjusts gradients according to input norm. Experiments demonstrate consistent improvements over RMSNorm, LayerNorm, and DyT across both language modeling (dense and MoE LLMs) and computer vision (ViT, ConvNeXt, DiT, MAE) tasks, with faster convergence and better zero-shot performance achieved with negligible additional parameters and minimal computational overhead.

**Strengths:**

The paper is technically detailed and the motivation is clear: existing normalization layers such as RMSNorm provide stability but lose information about the input magnitude. SeeDNorm offers a straightforward extension that dynamically rescales activations conditioned on the current input, improving adaptability to data variability and distributional shifts. The theoretical analysis is comprehensive, covering forward and backward derivations as well as scaling invariance, and helps clarify why the method remains stable. The experimental section is broad and systematic, spanning both language and vision tasks, and the results consistently show better convergence and downstream performance. Ablation studies on activation choice, initialization, weight decay, and the multi-head formulation strengthen the empirical support. The method is simple to integrate and performs reliably across different architectures.

**Weaknesses:**

Although the results are strong, the contribution may appear incremental because SeeDNorm can be viewed as a combination of RMSNorm and DyT that merges their strengths while mitigating DyT’s vanishing gradient issue. The proposed dynamic rescaling mechanism resembles existing modulation approaches such as gating or FiLM-style conditioning, differing mainly in how the rescaling term is computed. The theoretical analysis mainly focuses on stability and gradient behavior but gives limited insight into why the dynamic adaptation improves representation learning or generalization. The paper states that the computational overhead is negligible, but there is no quantitative evidence such as runtime or FLOPs comparison. In addition, the multi-head version is only applied to vision tasks, and it is unclear whether a similar strategy could benefit language models or whether it was found unnecessary.

**Questions:**

- How do $\alpha$, $\beta$, and $\gamma$ evolve during training? Do they converge to stable values or continue adapting dynamically?
- In Figure 3 (fourth subplot), SeeDNorm shows a temporary spike in loss; is this caused by instability in the rescaling term or by random training noise?
- Why is the multi-head variant used only for vision models? Would multi-head rescaling help stabilize training in language models, or is it redundant in that setting?
- Can the authors provide quantitative measures of computational overhead, such as runtime or FLOPs, to support the efficiency claim?
- How does SeeDNorm interact with normalization-free or re-scaling transformer variants, and could it complement them?

---

> ### Author Response · Authors · 2025-11-25
> **Response to Reviewer HBqF (Part1)**
>
> We sincerely appreciate your careful reading and constructive feedback, and we are greatly encouraged by your positive assessment of our work. Below are our responses, which we hope will satisfactorily address your concerns.
>
> ---
>
> > W1: The theoretical analysis mainly focuses on stability and gradient behavior but gives limited insight into why the dynamic adaptation improves representation learning or generalization.
> >
>
> `Response:` Thank you for your comment. In terms of mathematical formulation, we have already provided an analysis in the paper: SeeDNorm preserves more data dependencies—specifically, the norm information of tokens—during both forward and backward passes of the network. Compared to RMSNorm/LayerNorm with fixed data-independent scaling coefficients $\pmb{\gamma}$, SeeDNorm allows for more flexible adjustments, thus enhancing the network’s expressive power. Additionally, we have visualized the inputs and outputs of different SeeDNorm layers, which are included in the last section of the appendix in our **revised paper**. Here, the x-axis denotes input values, the y-axis denotes output values, and different colors represent different token samples. It can be observed that compared to DyT and RMSNorm, the outputs of SeeDNorm show more distinct separation between different samples—instead of being clustered together, all samples are more uniformly distributed across the entire space, indicating that the features of different tokens are more discriminative.
>
> ---
>
> > W2: There is no quantitative evidence such as runtime or FLOPs comparison
> >
>
> `Response:` We sincerely appreciate your valuable feedback. As detailed in our unified response, **“Unified Response to the Computational Overhead and Runtime Efficiency of SeeDNorm,”** we have thoroughly analyzed the computational cost, the number of parameters and module‑level latency.
>
> ---
>
> > W3: The multi-head version is only applied to vision tasks, and it is unclear whether a similar strategy could benefit language models or whether it was found unnecessary.
> >
>
> `Response:`We primarily adopt the multihead variant of SeeDNorm for vision tasks especially ImageNet image classification, where models are trained for many epochs on relatively small datasets and are therefore highly prone to overfitting. Overfitting often leads to increased gradient variance, which can affect training stability. In Section F of the appendix **“MORE ABLATION STUDIES,”** we also applied the multi‑head SeeDNorm to OLMoE. As shown in Table 18, this leads to negative effects, because training LLM involves sufficiently rich corpus and is far less susceptible to overfitting. In addition, maintaining a certain level of gradient variance is beneficial for training dynamically activated mixture‑of‑experts models.
>
> ---
>
> > Q1: How do $\pmb{\alpha}$, $\pmb{\beta}$ and $\pmb{\gamma}$ evolve during training? Do they converge to stable values or continue adapting dynamically?
> >
>
> `Response:`The parameters $\pmb{\alpha}$, $\pmb{\beta}$, and $\pmb{\gamma}$ continue to converge, although their trajectories and magnitudes differ across layers. For vision tasks, where training is already sufficiently thorough,  $\pmb{\alpha}$, $\pmb{\beta}$, and $\pmb{\gamma}$ have largely stabilized. In language modeling tasks, the rates of change for these parameters are steadily decreasing—indicating that they are approaching stability—though they are still in the process of converging.
>
> ---
>
> > Q2: In Figure 3 (fourth subplot), SeeDNorm shows a temporary spike in loss; is this caused by instability in the rescaling term or by random training noise?
> >
>
> `Response:` Thank you very much for your question and careful observation. During training, we record the changes in $\pmb{\alpha}$, $\pmb{\beta}$, and $\pmb{\gamma}$ parameters for each SeeDNorm Layer, and no spikes are observed in the updates of these parameters. We tend to believe this phenomenon is caused by input noise: first, the random masking of image patches in Masked Autoencoder (MAE) tasks itself leads to fluctuations in the input distribution; second, the initialization of mask tokens in MAE is inherently noisy. Thus, occasional spikes may appear in the early training stage, but they stabilize in the later stages.

---

> ### Author Response · Authors · 2025-11-25
> **Response to Reviewer HBqF (Part2)**
>
> ---
>
> > Q3: Why is the multi-head variant used only for vision models? Would multi-head rescaling help stabilize training in language models, or is it redundant in that setting?
> >
>
> `Response:`We have already answered this question in our response to `W3`.
>
> ---
>
> > Q4: Can the authors provide quantitative measures of computational overhead, such as runtime or FLOPs, to support the efficiency claim?
> >
>
> `Response:` As detailed in our unified response, **“Unified Response to the Computational Overhead and Runtime Efficiency of SeeDNorm,”** we have thoroughly analyzed the computational cost, the number of parameters and module‑level latency.
>
> ---
>
> > Q5: How does SeeDNorm interact with normalization-free or re-scaling transformer variants, and could it complement them?
> >
>
> `Response:` Thank you for your interest in the broader applicability of SeeDNorm. First, we still tend to believe that normalization layers remain essential components for training modern Transformers. Prior attempts to replace LN/RMSNorm with dyt show significantly worse performance in our OLMo and OLMoE experiments, and the accompanying theoretical analysis also indicates a potential risk of gradient vanishing. There are also some works that attempt to remove normalization entirely, such as T‑FixUp [1], but they have not been validated on modern decoder‑only LLMs.
>
> We are actually not very clear about the meaning of "interact". If it refers to applying SeeDNorm in these normalization-free Transformers, the SeeDNorm Layer can be inserted directly just like in classical Transformers. Meanwhile, we believe the idea of SeeDNorm can also be applied to different components—retaining more input-related information during the forward or backward pass of the network often further enhances the network’s expressive power. For example, in our response to Reviewer `NMms` `W3`, introducing the dynamic term of SeeDNorm into the scaling coefficient of GroupNorm layers still yielded benefits. However, we have not conducted further theoretical analysis; this is just an experiment provided for reference only, and we are also very interested in further exploring its application potential in other modules.
>
> [1] Huang X S, Perez F, Ba J, et al. Improving transformer optimization through better initialization[C]//International Conference on Machine Learning. PMLR, 2020: 4475-4483.

---

### Official Review · Reviewer_EBT8 · 2025-11-01

**Soundness:** 3
**Presentation:** 3
**Contribution:** 2
**Rating:** 6
**Confidence:** 3

**Summary:**

The authors point out that existing normalization layers, such as RMSNorm and LayerNorm, overlook the norm and scale information of the input. In situations where there are distributional shifts, the norms of the input can provide valuable information and a consistent scaling factor. The authors argue that ignoring this aspect reduces the expressiveness of models, particularly in large language models and vision models that operate across different domains. To address this issue, the proposed SeeDNorm introduces a dynamic scaling factor for normalization that is based on the input itself. This approach helps to retain some norm and scale information while adapting to each individual input sample. When training large-scale transformers, whether for language or vision tasks, and when facing varying input distributions, it may be beneficial to consider this proposed method as it could potentially enhance performance by avoiding the limitations of fixed normalization.

**Strengths:**

1. SeeDNorm offers improved preservation of input norm information and adaptability to shifts in input distribution. Unlike RMSNorm, which disregards input norms, SeeDNorm retains some of the "scale" information through a dynamic term. This scaling, being dependent on the input, allows it to better manage varying magnitudes or changes in the domain.

2. Experiments conducted across language tasks demonstrate that SeeDNorm accelerates convergence and produces better final results compared to the baseline methods of RMSNorm and LayerNorm.

3. The paper not only presents empirical results but also includes the derivation of gradients, discusses invariance properties, and addresses initialization and regularization choices for stability.

**Weaknesses:**

1. For extremely large models or those with many layers of normalization, the benefits of SeeDNorm may be limited. The paper indicates that the improvements diminish for dense models compared to Mixture of Experts (MoE) models.

2. Although SeeDNorm shows consistent improvement, the gains are often subtle, typically just a few tenths of a percent in accuracy, particularly for dense models, as illustrated in Table 2.

3. While the paper broadly discusses "vision tasks," it does not assess tasks that heavily involve spatial features, such as detection, oriented bounding boxes (OBB), and segmentation. More targeted experiments would strengthen the validity of their claims.

**Questions:**

1. How beneficial would SeeDNorm be in smaller models or resource-constrained environments, such as mobile or edge devices where memory and speed are crucial? Could the authors provide some insights on this?
2. How stable and easy is it to integrate SeeDNorm into different network architectures?
3. Does SeeDNorm generalize across various tasks without requiring extensive tuning?

---

> ### Author Response · Authors · 2025-11-25
> **Response to Reviewer EBT8 (Part1)**
>
> We sincerely thank the reviewer for the detailed and insightful feedback. Below are our responses, which we hope will satisfactorily address your concerns.
>
> ---
>
> > W1: For extremely large models or those with many layers of normalization, the benefits of SeeDNorm may be limited.
> >
>
> `Response:` The results in Figure 1 and Figure 2 of main text both show that as the number of training tokens increases, the loss reduction benefit of SeeDNorm continues to grow. In LLM settings, the adequacy of training can be assessed using the Token-Per-Parameter (TPP) metric. This implies that OLMoE‑7B needs to be trained with roughly seven times more tokens than OLMoE‑1B to reach a comparable level of training sufficiency. Due to time limitations at the time of submission, we trained the OLMoE‑7B model on only 1000B tokens, so the performance gains were a little less than OLMoE-1B. We have since continued training the 7B model to 2338B tokens (limited by time and resources, we didn't train more), and its results are shown in the table below.
>
> | **Model** | **Training Tokens** | **Training Loss** | **c4_en_validation Loss** |
> | --- | --- | --- | --- |
> | OLMoE-7B | 2338 B | 2.231 | 2.599 |
> | OLMoE-7B-SeeDNorm | 2338 B | 2.212 | 2.584 |
> | $\Delta$ |  | **-0.019** | **-0.015** |
>
> The results indicate that under a more comparable TPP level, OLMoE-7B still achieves a loss reduction benefit that is nearly consistent with that of OLMoE-1B. Additionally, since the baseline loss of OLMoE-7B itself is already significantly lower than that of OLMoE-1B, achieving a comparable relative loss reduction on such a low-loss, larger-scale model architecture is even more noteworthy. This observation confirms that the effectiveness of SeeDNorm can be maintained stably when scaled to larger LLM architectures.
>
> ---
>
> > W1: The paper indicates that the improvements diminish for dense models compared to Mixture of Experts (MoE) models.
> >
>
> `Response:` Firstly, we would like to note that MoE architectures have been adopted by the vast majority of recent large language models, so demonstrating strong performance in MoE settings is particularly relevant in practice.
>
> In addition, SeeDNorm is indeed more naturally suited to MoE structures compared with dense models. We believe this is because MoE routing is dynamic—meaning the same token may be sent to different experts—so the input distribution to the normalization layer can change in real time. The input‑dependent scaling mechanism in SeeDNorm can adapt to these variations.
>
> Furthermore, different experts in MoE models typically produce outputs with heterogeneous distributions (e.g., one expert may specialize in semantic tokens while another focuses on logical reasoning tokens). After passing through SeeDNorm, the dynamic scaling term can adjust based on each token’s specific input features, making subsequent processing easier. In contrast, dense models have more uniform token distributions, so the benefit of such dynamic adaptation is naturally more limited.
>
> ---
>
> > W2: Although SeeDNorm shows consistent improvement, the gains are often subtle, typically just a few tenths of a percent in accuracy, particularly for dense models, as illustrated in Table 2.
> >
>
> `Response:` We sincerely appreciate your comment, and we would like to respectfully clarify that many of the observed improvements are in fact not subtle. The experimental results presented in Table 2 are obtained from training and testing on ImageNet-1K. It is widely acknowledged that **further performance improvements on ImageNet-1K are inherently limited** due to long-standing issues with the dataset itself, such as annotation errors, class imbalance, and variations in sample quality. For example, compared to Swin Transformer Small [1], Swin Transformer Base  increases the parameter count by 66% and the computational overhead by nearly 80%, yet it only achieves a marginal 0.5% improvement in Top-1 accuracy on ImageNet-1K. In contrast, our proposed SeeDNorm introduces **negligible parameter increment and computational overhead (as detailed in our unified response “Unified Response to the Computational Overhead and Runtime Efficiency of SeeDNorm” to the reviewers),** With this near-zero additional cost, we directly achieved a Top-1 accuracy improvement of approximately 0.5% for both ViT-B and ViT-L models on ImageNet-1K. Considering the strict resource constraints of this "near-free lunch" setting, we believe this improvement could not be deemed subtle.
>
> Moreover, as shown in Table 1, our method yields substantial gains on language modeling tasks and improves many downstream tasks by several percentage points. Considering that these improvements come at almost no additional cost, we believe the benefits are meaningful and practically valuable.
>
> [1] Liu Z, Lin Y, Cao Y, et al. Swin transformer: Hierarchical vision transformer using shifted windows[C]//Proceedings of the IEEE/CVF international conference on computer vision. 2021: 10012-10022.

---

> ### Author Response · Authors · 2025-11-25
> **Response to Reviewer EBT8 (Part2)**
>
> ---
>
> > W3: While the paper broadly discusses "vision tasks," it does not assess tasks that heavily involve spatial features, such as detection, oriented bounding boxes (OBB), and segmentation. More targeted experiments would strengthen the validity of their claims.
> >
>
> `Response:` Our proposed SeeDNorm is primarily designed to be applied to **the currently dominant Transformer architectures** as well as network structures that use token-wise LayerNorm or RMSNorm, such as ConvNeXt.
>
> In detection/segmentation tasks, although some methods may adopt Transformer-based backbones, components like Feature Pyramid Networks (FPNs), Region Proposal Networks (RPNs), and prediction heads typically incorporate a large number of BatchNorm (BN) layers. Since BN operates along a completely different dimension (batch dimension) compared to LN/RMSNorm (token/feature dimension in individual samples), the improvements and theoretical analysis derived for SeeDNorm are not directly applicable to BN. This is precisely the reason why we did not attempt to verify SeeDNorm on detection/segmentation tasks in the original manuscript.
>
> However, we have now conducted exploratory experiments on these tasks. As shown in the following table, we selected MIMDet [2] as the baseline and **retrained three variants** of it: one with the original LayerNorm-based backbone, one with a DyT-based backbone, and one with a SeeDNorm-based backbone. All models are initialized using MAE pre‑training. SeeDNorm-based backbone architecture follows the MAE design described in Table 10 of our paper, and all other configurations remain consistent with those used in MIMDet.
>
> The results below indicate that MIMDet equipped with a SeeDNorm-based backbone achieves the optimal performance across all evaluated metrics.
>
> | **Model Variants** | **Detection Bbox AP** | **Segmentation Mask AP** |
> | --- | --- | --- |
> | LayerNorm (baseline) | 51.29 | 45.75 |
> | DyT | 50.72 | 45.13 |
> | **SeeDNorm** | **51.67** | **45.99** |
>
> As previously noted, we only replaced the LayerNorm layers within the backbone network. After feature extraction, backbone features are fused in the FPN and fed into subsequent prediction pipelines, where we retained all original BatchNorm layers without modification. This limitation introduces a certain bottleneck on the potential performance improvement. However, the fact that **performance gains can still be achieved by only modifying the backbone,** which demonstrates the effectiveness of SeeDNorm in detection/segmentation tasks.
>
> ---
>
> > Q1: How beneficial would SeeDNorm be in smaller models or resource-constrained environments, such as mobile or edge devices where memory and speed are crucial? Could the authors provide some insights on this?
> >
>
> `Response:` As outlined in our unified response to all reviewers in **“Unified Response to the Computational Overhead and Runtime Efficiency of SeeDNorm”**, the additional computational and parameter overhead introduced by our method is extremely small—almost negligible—and can be regarded as essentially a “free lunch.” Therefore, even in resource‑constrained settings, SeeDNorm can be reliably used as a drop‑in replacement for LayerNorm or RMSNorm.
>
> ---
>
> > Q2: How stable and easy is it to integrate SeeDNorm into different network architectures?
> >
>
> `Response:` Integrating SeeDNorm into existing network architectures is extremely straightforward: one only needs to **directly replace all token-wise normalization layers (such as LayerNorm/RMSNorm) in the network with SeeDNorm**, without requiring any other structural modifications.
>
> Regarding stability, we have already provided a detailed discussion in the main paper. For large language models, a direct substitution with the standard version of SeeDNorm is sufficient. For tasks such as ImageNet classification, where models are trained for many epochs on relatively small datasets and thus face a higher risk of overfitting, the multi‑head variant of SeeDNorm can be used to ensure stability.
>
>
> [2] Fang Y, Yang S, Wang S, et al. Unleashing vanilla vision transformer with masked image modeling for object detection[C]//Proceedings of the IEEE/CVF International Conference on Computer Vision. 2023: 6244-6253.

---

> ### Author Response · Authors · 2025-11-25
> **Response to Reviewer EBT8 (Part3)**
>
> ---
>
> > Q3: Does SeeDNorm generalize across various tasks without requiring extensive tuning?
> >
>
> `Response:` Yes, except for tasks with a relatively high risk of overfitting (as mentioned earlier, which require the use of the multi-head variant of SeeDNorm), SeeDNorm can be directly applied to diverse tasks without hyperparameter tuning. We maintain consistency in all hyperparameters across our experiments as reported in the main manuscript: for instance, the parameter $\pmb{\alpha}$  is uniformly initialized to 1, and the weight decay applied to SeeDNorm is kept identical to that of the overall network. With this consistent configuration, SeeDNorm delivered performance gains across all tasks evaluated in our paper. We acknowledge that these settings may not represent the optimal hyperparameters for every task; however, the fact that SeeDNorm performs well under a unified configuration further demonstrates its broad applicability.

---

### Author Response · Authors · 2025-11-25
**Unified Response to the Computational Overhead and Runtime Efficiency of SeeDNorm (Part1)**

We thank all reviewers for their insightful comments. We are greatly encouraged by the positive feedback provided. Since reviewers `EBT8`, `HBqF` both raise questions regarding the runtime efficiency of SeeDNorm, we provide here a unified response to these concerns.

### **Theoretical Analysis**

Compared to RMSNorm, SeeDNorm introduces two additional $D$-dimensional parameters, $\pmb{\alpha}$ and $\pmb{\beta}$. In the entire Transformer network, assuming each layer includes input normalization for the attention layer and FFN, QKNorm within the attention layer, and normalization at the Transformer output, the newly introduced parameters amount to $(2\times2D + 2\times2\times D/H) \times N + 2D = (4N+4\frac{N}{H}+2) \times D$, where $H$ is the number of attention heads. Since $N$ is much smaller than $D$, the increase in the overall parameter is much smaller than a linear layer, which can be considered negligible.

In terms of computational complexity, compared to RMSNorm, SeeDNorm introduces two additional matrix multiplications, one element-wise activation, and one element-wise addition along the channel dimension. For each token $\pmb{x} \in \mathbb{R}^{1\times D}$, compared to RMSNorm, the number of additional multiplications is $2D$, and the number of additional additions is $D+D-1$. The total additional multiply-add operations for the entire network are $(4D-1)\times (2N+1) +(\frac{4D}{H}-1)\times 2N = (8N+\frac{8N}{H}+4)\times D-4N-1=O(D)$. In contrast, a $D\times D$ linear layer already involves a computational complexity of $O(D^2)$. Therefore, the additional computational overhead introduced by SeeDNorm remains negligible.

### **Overall Running Efficiency**

Our theoretical analysis shows that the additional computation and parameters introduced by SeeDNorm are extremely small—essentially negligible—and can be regarded as almost a “free lunch.” We also report in the table below the changes in parameter count, computational cost, and tokens per second before and after integrating SeeDNorm into OLMoE‑1B and OLMoE‑7B during training.

| **Model** | **OLMoE-1B-SeeDNorm** | **OLMoE-1B** | **OLMoE-7B-SeeDNorm** | **OLMoE-7B** |
| --- | --- | --- | --- | --- |
| #Params | 1,310,670,336 | 1,310,639,104 | 6,919,243,776 | 6,919,161,856 |
| FLOPs Per Token (G) | 0.846511 | 0.846505 | 2.252853 | 2.252845 |
| Tokens Per Second Per GPU | 32754.78 | 35124.27 | 9778.12 | 9522.39 |

As shown, introducing SeeDNorm results in virtually no additional resource consumption. The tokens‑per‑second throughput is even slightly higher than the baseline in 7B model when SeeDNorm is enabled. This is because runtime speed can also be affected by factors such as I/O bandwidth and incidental CPU usage, so this metric should be interpreted with caution. Nonetheless, it still demonstrates that SeeDNorm has almost no impact on efficiency.

---

### Author Response · Authors · 2025-11-25
**Unified Response to the Computational Overhead and Runtime Efficiency of SeeDNorm (Part 2)**

### **Module-Level Standalone Running Efficiency**

The overall training speed of a model can be influenced by many external factors, such as I/O and CPU usage, which limits the interpretability of this metric. Moreover, since normalization layers inherently account for only a very small portion of the total computation, their impact on end‑to‑end training speed is minimal. Therefore, we additionally conducted a runtime efficiency analysis focused solely on the normalization module itself. In the table below, we report the forward‑pass latency of the SeeDNorm module compared with the RMSNorm module. The values represent the average latency over 1000 forward passes, excluding the first 10 iterations, since PyTorch may compile certain operators during initial execution.

As indicated in the table below, if implemented purely with PyTorch, SeeDNorm exhibits higher latency than RMSNorm. This is because a vanilla PyTorch implementation involves relatively more frequent global memory reads and writes, leading to non-negligible memory access overhead.

Therefore, we reimplemented the forward inference kernel of SeeDNorm using Triton (the corresponding source code is attached in the Appendix).

```python
@triton.jit
def seednorm_fwd_kernel(
    X, Y, W, alpha, beta, stride_ml, stride_n, L, N, eps, BLOCK_SIZE: tl.constexpr,
):
    row = tl.program_id(0)
    batch = tl.program_id(1)

    base_idx = row * stride_ml + batch * stride_n
    Y += base_idx
    X += base_idx

    _rms = tl.zeros([BLOCK_SIZE], dtype=tl.float32)
    _dot_product = tl.zeros([BLOCK_SIZE], dtype=tl.float32)
    for off in range(0, N, BLOCK_SIZE):
        cols = off + tl.arange(0, BLOCK_SIZE)
        a = tl.load(X + cols, mask=cols < N, other=0.0).to(tl.float32)
        beta_element = tl.load(beta + cols, mask=cols < N).to(tl.float32)
        _rms += a * a
        _dot_product += a * beta_element

    rms = tl.sqrt(tl.sum(_rms) / N + eps)
    dot_product = tl.sum(_dot_product)
    neg_two_x = -2.0 * dot_product
    exp_neg_two_x = tl.exp(neg_two_x)
    dot_product = (1.0 - exp_neg_two_x) / (1.0 + exp_neg_two_x)

    for off in range(0, N, BLOCK_SIZE):
        cols = off + tl.arange(0, BLOCK_SIZE)
        mask = cols < N
        w = tl.load(W + cols, mask=mask)
        alpha_element = tl.load(alpha + cols, mask=mask)
        x = tl.load(X + cols, mask=mask, other=0.0).to(tl.float32)
        x_hat = x / rms
        y = x_hat * (w + alpha_element * dot_product)
        tl.store(Y + cols, y.to(X.dtype.element_ty), mask=mask)

class SeeDNorm(Module):
  def __init__(self, D, init):
    super().__init__()
    self.alpha = Parameter(ones(D) * init)
    self.beta = Parameter(zeros(D))
    self.gamma = Parameter(ones(D))

  def forward(x):
    y = torch.empty_like(x)
    M, L, N = x.shape
    grid = (M, L)
    seednorm_fwd_kernel[grid](
        x, y, self.gamma, self.alpha, self.beta, x.stride(0), x.stride(1), L, N, self.eps, BLOCK_SIZE=1024
    )
    return y
```

With this optimized Triton kernel, the forward inference speeds of SeeDNorm and RMSNorm become nearly identical, as shown below.

- forward with single token

| **Method** | **RMSNorm** |  | **SeeDNorm** |  |
| --- | --- | --- | --- | --- |
| Implementation | PyTorch | Triton | PyTorch | Triton |
| Latency(s) | 6.0492548072943464e-05 | 2.8293310606386513e-05 | 0.00018696578626986593 | 2.8592990929610096e-05 |
- forward with 1024 tokens

| **Method** | **RMSNorm** |  | **SeeDNorm** |  |
| --- | --- | --- | --- | --- |
| Implementation | PyTorch | Triton | PyTorch | Triton |
| Latency(s) | 6.427329935831949e-05 | 2.8332560759736225e-05 | 0.00019195878121536225 | 2.835589839378372e-05 |

---

### Author Response · Authors · 2025-12-04
**Summary of Author Response**

**Dear Area Chairs, Senior Area Chairs, Program Chairs, Reviewers:**

---

We fully understand that the recent technical issues with OpenReview have hindered the interactive discussion that normally takes place during the rebuttal phase. We would like to express our sincere gratitude to the Area Chairs and Program Chairs for their extraordinary efforts under these challenging circumstances, and to the reviewers for their time and for carefully reading our responses. Since only one reviewer participated in the discussion before the incident, we have summarized the key points from the original reviews alongside our responses below. We hope this overview will assist the Area Chairs and alleviate their workload during the final evaluation, and also enable the Reviewers to quickly grasp the essence of our overall rebuttal.

We sincerely thank all reviewers for their positive assessments and for reaching a consensus that our work is **above the acceptance threshold**. We are deeply encouraged that all reviewers **unanimously recognized** the **soundness of our motivation and method, our comprehensive experiments, as well as thorough both empirical and theoretical analysis.**

In addition, we have provided **detailed point-by-point responses** to every weakness and question raised by each reviewer in our individual replies. Our rebuttal can be summarized as follows:

- **For Common Question on Computational Cost and Efficiency**
    - We address the concerns raised by multiple reviewers regarding computational cost and efficiency, we have conducted both **theoretical complexity analysis** and **empirical runtime tests**. Furthermore, we provided the **forward Triton Kernel implementation** for SeeDNorm, demonstrating that it incurs **negligible efficiency overhead**.
- **For Reviewer EBT8:**
    - We clarify that our method maintains **substantial gains** even when scaled to **larger models**. Furthermore, we provide an explanation for why the improvements are **more pronounced** on Mixture-of-Experts (MoE) architectures.
    - We clarify the **performance margins** on ImageNet, highlighting the **significant improvement** of our method.
    - We **further validate** the effectiveness of our method on **object detection and segmentation tasks**.
    - We elaborate on the **efficiency** of our method, the **simplicity** of our method’s application, specifically underlining its **plug-and-play** compatibility with existing architectures and tasks.
- **For Reviewer HBqF:**
    - We provide **additional insights** into how SeeDNorm enhances the model's **representation capability**, supported by utilizing **visualization plots** to illustrate the transformation between inputs and outputs.
    - We conduct a **detailed computational complexity and runtime efficiency analysis.**
    - We discuss specific **experimental observations** and **settings**, as well as the **feasibility of integrating** our method with other model architectures.
- **For Reviewer gTBE:**
    - We provid **additional details** regarding the **experimental configurations** and design.
    - We clarify the **performance margins** on ImageNet, highlighting the **significant improvement** of our method.
    - We elaborate on the **simplicity** of our method’s application, specifically underlining its **plug-and-play** compatibility with existing architectures and tasks.
    - We **address** concerns regarding **reproducibility**, **parameter selection**, **paper written**, and correct typos.
- **For Reviewer NMms:**
    - We provide **further explanation** regarding the **models used in our experiments**.
    - We provide experiments **integrating** the **principle of SeeDNorm** with Group Normalization (GN), demonstrating the **broader application potential of SeeDNorm.**

We wish everyone a very pleasant day, and we hope our summary will be helpful to you.

---

Best regards,

Authors of Submission 4326

---

### Meta-Review · Area_Chair_KbZM · 2026-01-08

**Summary:**

This paper proposes SeeDNorm, a novel dynamic normalization layer that adaptively adjusts the scaling coefficient based on the current input to preserve input norm information.
It achieves superior performance across large language model pre-training and various computer vision tasks with negligible computational overhead and minimal additional parameters.
1. Reviewers EBT8 and gTBE pointed out that the performance gains on dense models and ImageNet are relatively subtle compared to Mixture-of-Experts (MoE) models.
2. Reviewers EBT8, HBqF, and gTBE expressed concerns regarding the computational cost, runtime efficiency, and the lack of quantitative evidence like FLOPs or latency comparisons.
3. Reviewer EBT8 noted that the paper lacked assessment on vision tasks involving heavy spatial features, such as object detection and segmentation.
4. Reviewer gTBE pointed out a potential typo in Theorem 3.2 regarding the relationship between dot products and dimensions, and questioned the reproducibility of experimental results.
5. Reviewer NMms questioned whether SeeDNorm could be combined with other normalization techniques like Group Normalization (GN).

Considering the strong empirical results across multiple domains and the thorough technical rebuttal that addressed all efficiency and theoretical concerns, I tend to accept this paper.

**Reviewer Concerns:**

1. The authors clarified that while gains on dense models are smaller, they are consistent and achieved with "near-zero" cost; they also provided new results for OLMoE-7B at higher TPP levels. I believe this addresses the concern by showing scalability.
2. The authors provided a detailed theoretical complexity analysis and empirical runtime tests using a custom Triton kernel, showing negligible overhead. This effectively resolves the efficiency concerns.
3. The authors conducted new exploratory experiments on MIMDet for detection and segmentation, showing that SeeDNorm-based backbones outperform LayerNorm and DyT baselines. This successfully validates the method's effectiveness on spatial tasks.
4. The authors corrected the typo in Theorem 3.2 and clarified that all LLM experiments used fixed, identical seeds for reproducibility. These clarifications are professional and resolve the technical doubts.
5. The authors implemented a "SeeD-GN" variant and provided preliminary results showing improvements on ResNet50. This demonstrates the broader potential of the proposed mechanism.

**Reviewer Scores:**

- Reviewer EBT8: 6 -> 8, Concerns 1 and 3 were well-addressed with new experimental evidence on larger models and detection tasks.
- Reviewer HBqF: 8 -> 8, the original score was already high, and concern 2 was perfectly resolved by the Triton kernel implementation and quantitative analysis.
- Reviewer gTBE: 6 -> 8, the author's response successfully corrected the theoretical typo in C4 and provided the requested experimental details.
- Reviewer NMms: 6 -> 8, the additional experiments on Group Normalization in concern 5 clearly demonstrated the generality of the method.

---

### Decision · Program_Chairs · 2026-01-26

Accept (Poster)